# Extreme rainstorms drive exceptional organic carbon export from forested humid-tropical rivers in Puerto Rico

K. E. Clark [1,6] ✉, R. F. Stallard[2], S. F. Murphy [2], M. A. Scholl[3], G. González [4], A. F. Plante [1] & W. H. McDowell [5]

Extreme rainfall events in the humid-tropical Luquillo Mountains, Puerto Rico export the bulk of suspended sediment and particulate organic carbon. Using 25 years of river carbon and suspended sediment data, which targeted hurricanes and other large rainstorms, we estimated biogenic particulate organic carbon yields of $65 \pm 16$ tC km$^{-2}$ yr$^{-1}$ for the Icacos and $17.7 \pm 5.1$ tC km$^{-2}$ yr$^{-1}$ for the Mameyes rivers. These granitic and volcaniclastic catchments function as substantial atmospheric carbon-dioxide sinks, largely through export of river biogenic particulate organic carbon during extreme rainstorms. Compared to other regions, these high biogenic particulate organic carbon yields are accompanied by lower suspended sediment yields. Accordingly, particulate organic carbon export from these catchments is underpredicted by previous yield relationships, which are derived mainly from catchments with easily erodible sedimentary rocks. Therefore, rivers that drain petrogenic-carbon-poor bedrock require separate accounting to estimate their contributions to the geological carbon cycle.

[1] Department of Earth and Environmental Science, University of Pennsylvania, Philadelphia, PA, USA. [2] U.S. Geological Survey, Water Resources Mission Area, Boulder, CO, USA. [3] U.S. Geological Survey, Water Resources Mission Area, Reston, VA, USA. [4] USDA Forest Service, International Institute of Tropical Forestry, Río Piedras, PR, USA. [5] Department of Natural Resources and the Environment, University of New Hampshire, Durham, NH, USA. [6] Present address: Department of Geography & Planning, University of Liverpool, Roxby Building, Liverpool L69 7ZT, United Kingdom. ✉email: Kasey.clark@liverpool.ac.uk

Rivers play multiple roles in the geological carbon cycle, with some catchments acting as net atmospheric carbon sinks (typically having carbon-poor and sulfide-poor bedrock) and others as net sources (abundant petrogenic organic carbon or sulfides)[1]. To properly understand the role of rivers in the global carbon cycle, we must assess river carbon budgets in different types of rivers with varying weathering rates, geology, and rainfall regimes. Biogenic organic carbon (OC) erosion is an integral component of the geological carbon cycle[2,3]. Each river catchment has its own net carbon budget, with various carbon (atmospheric $CO_2$) sources and sinks[2].

Igneous montane islands, such as Guadeloupe[4,5] and Hawaii[6], are distinct because these largely contain only carbon sinks. There is a drawdown of atmospheric $CO_2$ when river particulate organic carbon (POC) from the biosphere (biogenic POC) is buried in long-term ocean deposits and dissolved inorganic carbon (DIC) is produced during silicate weathering[2]. There is an effective release of $CO_2$ from rocks to the atmosphere when petrogenic organic carbon and sulfides in bedrock oxidize[2,7]. Carbon budgets of humid-tropical igneous montane islands and continental landscapes with island-arc geology are rarely assessed, and yet the erosion of biogenic POC from these catchments plays a potentially significant, yet poorly characterized, role in the geological carbon cycle.

River biogenic POC transport is tightly coupled with suspended sediment (SS) transport, with an established coupled exponential relationship among global rivers[3]. Using regressions, global biogenic POC fluxes are estimated based on known river SS yields[3]. As a result, the annual OC biogenic erosion is estimated at 110–230 MtC per year[3] and it has been included in the emerging view of the geological carbon cycle[2]. Most (86.5%) rivers in the global compilation drain sedimentary rocks rich in petrogenic carbon[3,8].

At the watershed scale, carbon transport is closely linked to both physical and chemical erosion[9]. The annual yields of dissolved carbon export, specifically dissolved organic carbon (DOC) and DIC, typically increase linearly with discharge[9]. River POC export, however, behaves nonlinearly, and up to 75% of the annual POC flux is exported during highest river discharge[5,10]. Yet despite the importance of extreme events to the export of sediment and POC, they are rarely measured in situ in rivers[9,10]. Because extreme rain events are projected to become more frequent and more severe in the tropics, quantifying present river POC export is particularly important, to establish a baseline, as POC fluxes will likely increase in the future[11].

In rivers with large SS yields that flow directly into the ocean, over 65% of river biogenic POC can be buried in offshore sedimentary deposits[2,12–14]. One way to assess the impact of river POC export from an ecosystem is to estimate the river biogenic POC export as a percentage of net primary productivity (NPP) – NPP$_{export}$[3]. In the tropics, landscape NPP and river POC fluxes are rarely quantified in the same catchment.

In this study, we assess river SS and river carbon (biogenic POC, DOC, DIC) during large to extreme rainfall events in humid-tropical, naturally vegetated catchments in eastern Puerto Rico that drain bedrock with little petrogenic carbon and sulfides. We assess SS and carbon yields from rivers, estimate biogenic POC$_{burial}$ in the nearby ocean, estimate carbon transfer within the geological carbon cycle, estimate the percentage of NPP$_{export}$ as POC, and assess river carbon fluxes during extreme rainfall events. We compare relationships between SS and biogenic POC in these rivers to global rivers.

## Results and discussion

### River particulate organic carbon in the Luquillo Mountains.
The two study rivers, Icacos and Mameyes, are in the Luquillo Mountains in Puerto Rico (See Supplementary Fig. 1), and are both small catchments (3.26 km² and 17.8 km², respectively) with granodioritic (Icacos) and volcaniclastic (Mameyes) bedrocks that do not contain substantial petrogenic organic carbon, reduced sulfur, or carbonates[9,15]. These catchments have very high rainfall (average of 5000 and 3720 mm yr$^{-1}$ in the Icacos and Mameyes, respectively)[16,17] and runoff (3906 and 2770 mm yr$^{-1}$, respectively) (Table 1). This study includes >4000 river samples collected over 25 years, distributed across four orders of magnitude of SS and POC concentrations and three orders of magnitude of runoff rates (Fig. 1). Using continuous river discharge measured by the US Geological Survey (USGS)[18,19], river constituent concentration data (See Supplementary Data 1), and the USGS program LOADEST[20], we determined river yields from 1991 to 2015.

River biogenic POC yields were 65 ± 16 tC km$^{-2}$ yr$^{-1}$ for the Icacos and 17.7 ± 5.1 tC km$^{-2}$ yr$^{-1}$ for the Mameyes (Table 1 and See Supplementary Data 2), which are significantly higher than previous reports for these rivers (3.8 tC km$^{-2}$ per year[21] and 22 tC km$^{-2}$ per year[9] for Icacos, 4 tC km$^{-2}$ per year[9] for Mameyes; See Supplementary Table 1). Previously reported river POC yields were underestimated because one study did not collect samples at the highest discharges[21] and in the other study[9,22] POC% was estimated from a relation with "loss on ignition" (LOI) of sediment samples [mass lost when baked at high temperature (550 °C), which includes organic matter plus water of hydration from clays[9]] and an average sediment/POC relation for global rivers[23] (See Supplementary Table 1) which is too low for these highly weathered, clay-rich catchments. In contrast to the previous work, this current study includes samples collected across a very wide range of runoff rates, and SS samples were analyzed for POC% with either an elemental analyzer or a carbon coulometer, rather than estimation from LOI.

The biogenic POC yields we obtained for Icacos and Mameyes are generally similar to yields from other montane rivers[8] (See Supplementary Data 3). The Capesterre River in Guadeloupe has

**Table 1 Mean annual yields for Icacos and Mameyes Rivers from 1991 to 2015, where ± is the 95% confidence interval. See Supplementary Data 2 for annual values.**

|  | Icacos | Mameyes |
|---|---|---|
| Area (km²) | 3.26 | 17.8 |
| Rainfall (mm yr$^{-1}$) | 5000 ± 482 | 3720 ± 359 |
| Runoff (mm yr$^{-1}$) | 3906 ± 339 | 2770 ± 227 |
| SS[a] (t km$^{-2}$ yr$^{-1}$) | 1530 ± 410 | 310 ± 96 |
| Biogenic POC (tC km$^{-2}$ yr$^{-1}$) | 65 ± 16 | 17.7 ± 5.1 |
| PN (tN km$^{-2}$ yr$^{-1}$) | 2.14 ± 0.52 | 0.70 ± 0.19 |
| DOC[a] (tC km$^{-2}$ yr$^{-1}$) | 12.4 ± 1.5 | 8.3 ± 0.9 |
| DIC[a] (tC km$^{-2}$ yr$^{-1}$) | 11.0 ± 0.5 | 13.4 ± 0.6 |
| Biogenic POC$_{burial}$[b] (tC km$^{-2}$ yr$^{-1}$) | 14.3$^{+7.7}_{-6.0}$ | 3.9$^{+2.3}_{-1.7}$ |
| Biogenic POC$_{burial}$[b] (tC yr$^{-1}$) | 46.6$^{+25.0}_{-19.6}$ | 69.3$^{+40.1}_{-31.1}$ |
| Carbon budget[c] (tC km$^{-2}$ yr$^{-1}$) | −25.3$^{+6.3}_{-7.9}$ | −17.3$^{+2.1}_{-2.6}$ |
| Carbon budget[c] (tC yr$^{-1}$) | −83.4$^{+20.6}_{-25.9}$ | −307.2$^{+38.1}_{-46.5}$ |
| Catchment NPP[d] (tC km$^{-2}$ yr$^{-1}$) | 990 ± 1 | 1013 ± 85 |
| NPP$_{export}$[e] (%) | 6.6 ± 1.6 | 1.7 ± 0.7 |
| Total river carbon[f] (tC km$^{-2}$ yr$^{-1}$) | 88 ± 18 | 39 ± 6 |
| Biogenic POC:DOC | 5.0 ± 1.0 | 2.0 ± 0.2 |
| NPP$_{export\_DOC}$[g] (%) | 1.3 ± 0.2 | 0.8 ± 0.2 |

[a]Yields updated from Stallard[9].
[b]Biogenic POC burial efficiency was estimated at 22 ± 5% based on the global average for terrestrial organic matter from Burdige[28].
[c]Carbon budget = Biogenic POC$_{burial}$ + DIC, the negative value represents an atmospheric $CO_2$ sink.
[d]Catchment NPP estimated from[45, 62], see Methods for details.
[e]NPP$_{export}$ (%) = Biogenic POC yield/catchment NPP × 100.
[f]Total river carbon = POC + DOC + DIC.
[g] NPP$_{export\_DOC}$ (%) = DOC yield / catchment NPP * 100

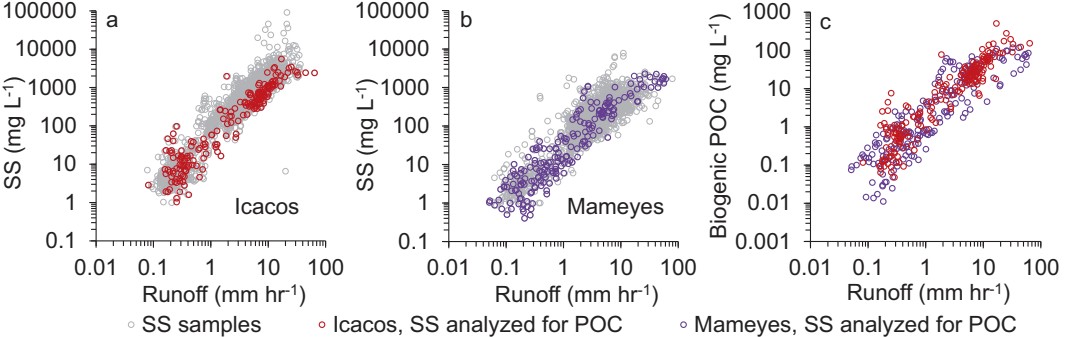

**Fig. 1 River constituent concentrations in relation to runoff in the Luquillo Mountains.** The relationship between river suspended sediment (SS mg L$^{-1}$) relative to river runoff (mm hr$^{-1}$) for **a** Icacos and **b** Mameyes, with SS samples analyzed for particulate organic carbon (POC)% represented by colored circles. **c** Relationship between measured biogenic POC (mg L$^{-1}$) relative to runoff. Biogenic POC (mg L$^{-1}$) was estimated from SS samples that were not measured for POC% (gray circles) using Eq. 5 for Icacos and Eq. 6 for Mameyes.

similar characteristics (geology, topography, vegetation, climate) and a POC yield of 18.3 tC km$^{-2}$ per year[5], comparable to $17.7 \pm 5.1$ tC km$^{-2}$ yr$^{-1}$ for Mameyes. In five of its 25 years of data, however, the Icacos had higher yields (97 to 179 tC km$^{-2}$ yr$^{-1}$) (See Supplementary Data 4) than the highest recorded biogenic POC yield of 87 tC km$^{-2}$ yr$^{-1}$ from Whataroa, New Zealand[24] (See Supplementary Data 3). These 5 years included numerous tropical storms and other high-intensity rainfall events that were largely captured by our storm sampling[25].

In the context of the geological/long-term carbon cycle, however, the burial of the exported biogenic POC is most relevant. Although there have not been any direct measurements of biogenic POC$_{burial}$ in the coastal regions of Puerto Rico, rivers discharge onto the shallow submarine shelf, where river POC spreads out and is reworked in the oxygenated environment[26]. It is expected that Icacos and Mameyes have a lower carbon burial efficiency because they have lower SS yields[2]. During extreme events, when over half of biogenic POC export occurs (See Supplementary Data 5), the POC is mainly derived from organic soil and vegetation, but may be converted back to carbon dioxide in the marine environment before deposition (~90% of organic matter in marine deposits is associated with minerals, such as clays and silts[27]). Without measurements available from the ocean adjacent to eastern Puerto Rico, we estimate biogenic POC$_{burial}$ using the global river estimates of $22 \pm 5$%[28]. The biogenic POC$_{burial}$ is estimated at $14.3^{+7.7}_{-6.0}$ and $3.9^{+2.3}_{-1.7}$ tC km$^{-2}$ yr$^{-1}$ for the Icacos and Mameyes, respectively (Table 1 and See Supplementary Data 2). The Icacos biogenic POC$_{burial}$ yields overlap with those from the Hokitika and Whataroa Rivers, New Zealand, and yields from the Mameyes overlap with the Peel and Arctic Red Rivers, in Canada, and the Capesterre River, in Guadeloupe[2].

The estimated geological carbon (atmospheric CO$_2$) sink was $-25.3^{+6.3}_{-7.9}$ and $-17.3^{+2.1}_{-2.6}$ tC km$^{-2}$ yr$^{-1}$ for the Icacos and Mameyes respectively (Table 1), which comprised POC$_{burial}$ and DIC production from silicate weathering. The largest known carbon sink by area is the Whataroa catchment in New Zealand, $-33 \pm 16$ tC km$^{-2}$ per year[2]; this range, however, overlaps the ranges of the Icacos and Mameyes. The dominant component of the carbon budget in the New Zealand catchment was a large biogenic POC$_{burial}$, despite a high carbon source contribution from petrogenic POC oxidation[2]. The Capesterre catchment in Guadeloupe is similar to the Icacos and Mameyes, with an estimated carbon sink of $-19 \pm 8$ tC km$^{-2}$ per year[4,5]. The carbon budgets in the Mameyes and Capesterre are dominated by chemical weathering of silicates (See Supplementary Data 2). This is not true for the Icacos, where physical erosion and subsequent POC$_{burial}$ may contribute over half of the carbon budget (See

Supplementary Data 2), despite the fact that it has one of the fastest chemical weathering rates of granitic rocks in the world[29]. To determine the true extent to which the Luquillo catchments are geological carbon sinks, a comprehensive assessment of river biogenic POC burial efficiency in the marine environment is needed.

**Biogenic POC and petrogenic-carbon-poor bedrock.** Global biogenic POC yields are $1.46^{+0.68}_{-0.47}$ tC km$^{-2}$ per year[3], as estimated from river SS yields using a regression developed by Galy, et al.[3]. Applying the Galy, et al.[3] regression to our study rivers, the biogenic POC yields are estimated at $4.7 \pm 1.7$ and $1.9 \pm 0.8$ tC km$^{-2}$ yr$^{-1}$ for Icacos and Mameyes, respectively. This underestimates actual river biogenic POC yields by $92 \pm 2$% and $87 \pm 3$%, respectively. It appears that in rivers with low to moderate SS yields but high biogenic POC yields, such as in the present study, the regression developed by Galy, et al.[3] may significantly underestimate biogenic POC yields (Fig. 2a). The higher biogenic POC yields relative to SS yields are caused by differences in geology, vegetation, and precipitation. The Luquillo catchments are underlain by igneous and volcaniclastic bedrock, which is less erodible than marine sediments[30], resulting in relatively lower SS yields, while large organic carbon stocks on the landscape are transported to rivers by extremely high precipitation.

There are distinct relationships between biogenic POC relative to SS yield in rivers with petrogenic-carbon-rich and petrogenic-carbon-poor (i.e., igneous and aluminosilicate metamorphic) bedrock (Fig. 2a). We propose that a separate regression be used for rivers draining bedrock that is petrogenic-carbon-poor (<2% petrogenic OC% of total POC). We selected this cut-off because it was a natural break in the global dataset (See Supplementary Data 3). These rivers have a different relationship between biogenic POC and SS, with a global mean $9.0 \pm 2.8$% biogenic POC ($n = 16$) (this study and refs. [2,3,5,6,31–34]); whereas rivers that drain bedrock that is petrogenic-carbon-rich have a global mean of $0.87 \pm 0.66$ biogenic POC% ($n = 84$)[2,3,35]. The biogenic POC% differences influence yields between the two groups (Fig. 2a). The proposed new regression for biogenic POC yield for rivers that have petrogenic-carbon-poor bedrock (Y$_{bio\_pp}$) is:

$$Y_{bio\_pp} = 0.100 \times Y_{SS}^{0.90}, \; r^2 = 0.96, \; n = 68 \qquad (1)$$

where Y$_{ss}$ represents the annual SS yield. The proposed new regression for petrogenic-carbon-rich bedrock (Y$_{bio\_pr}$) is:

$$Y_{bio\_pr} = 0.051 \times Y_{SS}^{0.64}, \; r^2 = 0.83, \; n = 84 \qquad (2)$$

More broadly speaking, biogenic POC export from rivers draining humid-tropical igneous mountainous islands and island

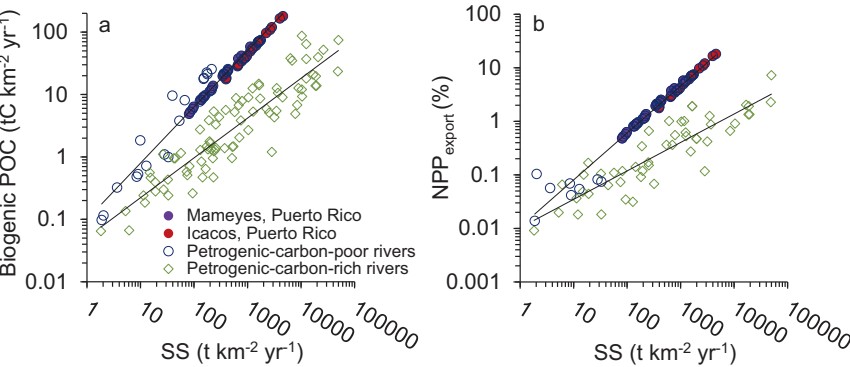

**Fig. 2 River suspended load relationships for global rivers.** The relationships are between **a** biogenic particulate organic carbon (POC) and **b** net primary productivity (NPP)$_{export}$ (%) relative to suspended sediment (SS) yield for global rivers.

arcs, which encompass a total area of 320,000 square km[5,36,37], are also likely underestimated. By assuming that these rivers are similar to our study rivers, we estimated biogenic POC fluxes, using the SS yields from Mameyes as the lower bound and from Icacos as the higher bound. We determined the biogenic POC yields using the regression developed by Galy, et al.[3] and then multiplied the area[5,36,37] to approximate biogenic POC fluxes for this physiographic setting. Based on the Galy, et al.[3] regression, humid-tropical igneous montane rivers produce 0.64 to 1.57 MtC yr$^{-1}$ of biogenic POC. On the other hand, following the same steps outlined above, but using Eq. 1, this setting would contribute fluxes of 5.59 to 23.52 MtC yr$^{-1}$. This represents a potentially substantial underestimation of biogenic OC export from humid-tropical igneous mountainous islands in the geological carbon cycle[2,3]. Accordingly, global biogenic OC export, which is currently estimated at 110–230 MtC per year[2], may be underestimated by 2.1 to 8.7% yr$^{-1}$. As more research is conducted in these settings, their relative importance to global biogenic POC export will become more apparent.

Both Luquillo rivers had NPP$_{export}$ by biogenic POC in the upper range reported for global rivers (Icacos 6.6 ± 1.6%, Mameyes 1.7 ± 0.7% (See Supplementary Data 4, 6). For almost 1/4 of the years, the Icacos had a greater export than the highest published NPP$_{export}$ (See Supplementary Data 4), Peinan, Taiwan, at 7.26% NPP$_{export}$[3] (See Supplementary Data 6). This Taiwanese river and the Icacos River had similar percent NPP$_{export}$, where both catchments had very high biogenic POC yields and a higher-than-average catchment NPP (See Supplementary Data 6). The high NPP$_{export}$ for the Icacos is plausible as physical erosion is in excess of equilibrium with chemical erosion[9]. Thus, soils that developed over thousands of years may now be eroding faster than they are developing[9].

While much less is known about NPP$_{export}$ (%), the global relationship derived by Galy, et al.[3] was largely derived from rivers with petrogenic-carbon-rich bedrock. The new regression for NPP$_{export}$ for rivers that have petrogenic-carbon-poor bedrock (NPP$_{export\_pp}$) is:

$$NPP_{export\_pp} = 0.011 \times Y_{SS}^{0.88}, r^2 = 0.96, n = 58 \quad (3)$$

and that for petrogenic-carbon-rich bedrock (NPP$_{export\_pr}$) is:

$$NPP_{export\_pr} = 0.010 \times Y_{SS}^{0.53}, r^2 = 0.78, n = 50 \quad (4)$$

Following the regression developed by Galy, et al.[3], the Luquillo rivers are significantly underestimated, with NPP$_{export}$ estimates of 0.48 ± 0.06% and 0.22 ± 0.03% from the Icacos and Mameyes, respectively. Although erosion rates are very high in the Icacos catchment relative to other watersheds in eastern Puerto Rico[9,38], SS yields are only considered moderate when

compared to global montane rivers with sedimentary bedrock[2]; however, the Icacos has one of the highest estimated percentages of NPP$_{export}$ by POC in the world, suggesting that separate relationships exist between these two bedrock types (Fig. 2b).

**River carbon fluxes during extreme rainfall events.** Extreme rainfall events (at our research site, >92 mm/event, which is within the top 1% of events by amount, with a mean intensity of 5.8 mm/hr) deliver ~18% of annual rainfall and generate 13% of annual runoff at the Luquillo rivers, yet export 54–63% of annual SS and 52–60% of annual biogenic POC yields (Fig. 3 and See Supplementary Data 5). These extreme events are responsible for approximately half of the annual biogenic POC export. While the DIC yields are similar in both catchments, over 80% of export occurs during small rainfall events (<22 mm per event) and dry periods (See Supplementary Data 5). DOC export closely follows rainfall amounts, rather than river runoff, where about half the annual rainfall and DOC export takes place during small rainfall events (<22 mm per event) and dry periods (Fig. 3 and Supplementary Data 5).

Annual biogenic POC and DOC yields are both higher in the Icacos than in the Mameyes. The biogenic POC:DOC for the Mameyes, 2.0 ± 0.2, is similar to that of the Capesterre catchment in Guadeloupe at 2.5[4,5], whereas the Icacos has a ratio of 5.0 ± 1.0, indicating dominance of particulates in river OC export (See Supplementary Data 2). The ratio increases during extreme rainfall events and decreases during small rainfall events (<22 mm per event) and dry periods (See Supplementary Data 5). Almost twice as much NPP$_{export}$ (%) as DOC (NPP$_{export\_DOC}$) is exported from the Icacos than from the Mameyes, whereas in the case of NPP$_{export}$ (%) as POC the Icacos is more than three times larger than from the Mameyes (Table 1 and See Supplementary Data 2).

Our yield estimates span 25 years (1991–2015) in Puerto Rico and included hurricanes Hortense (1996) and Georges (1998), which were in the top three years of river suspended load exports (See Supplementary Data 2); however, the longer-term river export due to hurricanes is likely much higher given that the largest recorded hurricanes, San Ciprian (1932), Hugo (1989), Irma (2017), and Maria (2017), occurred outside of this study period[39–41]. The significance of extreme-event-driven export will likely increase, as the Intergovernmental Panel on Climate Change (IPCC) projects a reduction in mean annual precipitation for the Caribbean and Central America, but with more extreme precipitation, occurring during tropical cyclones[11].

## Methods

**Study area.** The Luquillo Mountains of Puerto Rico have high rates of precipitation, physical erosion, hillslope turnover, and chemical weathering[9,15,29,38,42,43]. Landcover is mostly humid-tropical forest with high net primary productivity

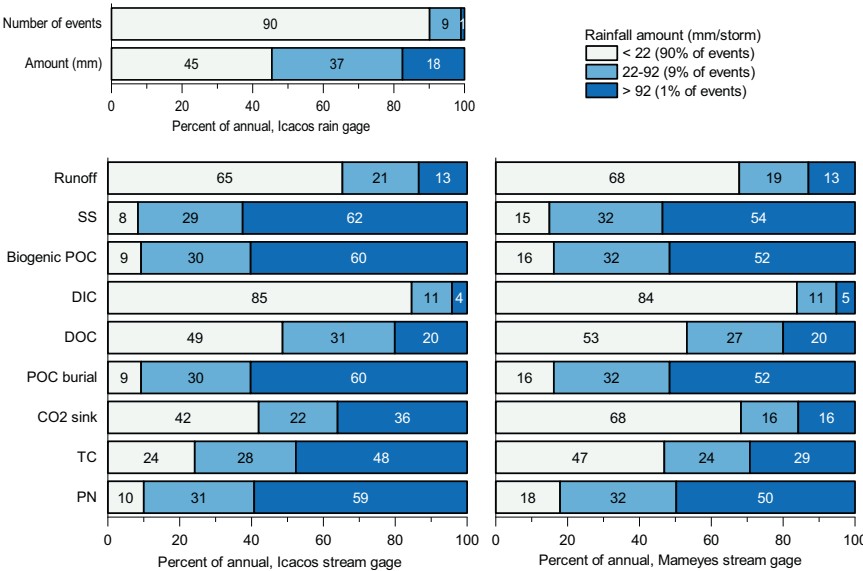

**Fig. 3 Percent contribution of annual runoff and river exports by rainfall event size for the Icacos and Mameyes Rivers.** These exports are separated by rainfall amount as a percentage of annual total (SS suspended sediment, POC biogenic particulate organic carbon, DIC dissolved inorganic carbon, DOC dissolved organic carbon, POC burial estimated POC burial in ocean if biogenic POC$_{burial}$ efficiency is estimated at 22%[28], CO2 sink estimated atmospheric carbon sink (POC$_{burial}$ + DIC), TC total carbon (POC + DIC + DOC), PN particulate nitrogen).

(NPP)[15,44–46]. The Luquillo Experimental Forest (LEF), coterminous with El Yunque National Forest, has been the site of decades of scientific investigations[47–50], including the National Science Foundation-funded Luquillo Long-Term Ecological Research (LTER) program and Luquillo Critical Zone Observatory (LCZO) program, and the U.S. Geological Survey (USGS)-funded Water, Energy, and Biogeochemical Budgets (WEBB) program. Our two-catchment study is in the LEF. The Icacos catchment (USGS station 50075000 Río Icacos near Naguabo, PR)[19] covers 3.26 km$^2$ and has an elevation of 620 to 830 m above sea level (masl) (See Supplementary Fig. 1). The Mameyes catchment (USGS station 50065500 Río Mameyes near Sabana, PR)[18] covers 17.8 km$^2$ with an elevation of 80 to 1050 masl[15] (See Supplementary Fig. 1). The mean-maximum-annual temperature was 26.0 °C and the mean-minimum-annual temperature was 20.8 °C at the El Verde Field station located in El Yunque Forest[51]. Average annual rainfall (from 1991 to 2015) was 5000 mm yr$^{-1}$ in Icacos and 3720 mm yr$^{-1}$ in Mameyes[16,17] (Table 1 and See Supplementary Data 2). The two catchments are described in detail in Murphy and Stallard[25].

The geology of the Icacos catchment consists of coarse-grained granitic rock that weathers to quartz and clay-rich, sandy soils[15]. The Mameyes catchment contains a portion of granitic rock, but the majority consists of fine-grained volcaniclastic rocks that weather to quartz-poor clayey soils[15]. The sources of suspended sediment erosion from the study catchments are from surficial erosion, bed and bank erosion, and from landslides[38]. The total hillslope erosion rate is estimated to be 750 and 523–2143 t km$^{-2}$ yr$^{-1}$ for the Icacos and Mameyes catchments, respectively; the Mameyes had a particularly high prehistoric landslide erosion rate (2000 t km$^{-2}$ yr$^{-1}$)[38]. Chemical weathering rates of silicate rocks in the Icacos are among the fastest in the world[29], with a weathering front of approximately 1 cm per 100 years[52]. The volcaniclastic rock of the Mameyes weathers faster, and both catchments have a thick layer of saprolite[29,43].

**River discharge and suspended sediment sampling.** Discharge was determined at the Icacos and Mameyes stream gaging stations maintained by the USGS[18,19]. A gap-free and evenly spaced instantaneous river discharge dataset for both rivers was used in order to determine annual water budgets and model constituent loads[22].

Through two distinct efforts, river SS and solute samples were collected at both river gages from 1991 to 2015 (See Supplementary Data 1). River samples were collected from 1991 to 2015 by the USGS WEBB program[53,54], with a focus on sampling during large rainfall events from 1992 to 2004 (See Supplementary Fig. 2 and Supplementary Data 1). Samples were collected across the hydrograph through a combination of manual sampling below the water surface as far from the river banks as was safely possible ($n = 336$), depth-integrated sampling ($n = 272$), and event sampling ($n = 3391$) using a sample port anchored below the water surface one meter from the riverbank using a portable sampler (1-liter bottles) (Teledyne ISCO 6712)[55] (See Supplementary Fig. 2 and Supplementary Data 1). Water samples were filtered for suspended and dissolved load with a 0.2-µm filter membrane[55]. During the USGS WEBB program, far more samples of most dissolved and suspended constituents were collected than in other studies. Most importantly, river SS samples analyzed for biogenic POC were collected during large rainfall events, including extreme rainfall events where runoff was >10 mm hr$^{-1}$ at both Icacos ($n = 1$) and Mameyes ($n = 7$) (See Supplementary Data 1). There have only been two other rivers where SS samples analyzed for biogenic POC were collected at extremely high runoff, specifically 11.2 mm hr$^{-1}$ in Taiwan[13] and 11.1 mm hr$^{-1}$ in the Erlenbach in the Swiss Alps[56].

River SS samples from the Icacos and Mameyes were also collected by the Lower Río Mameyes Water Quality Project. This was a collaborative effort between the USDA Forest Service International Institute of Tropical Forestry, the University of Puerto Rico, Luquillo LTER, and Luquillo CZO (LCZO). Sampling sites were the same as for the WEBB Project. Manual samples were collected from 1997 to 2015, where samples were taken away from the bank and below the water surface. Water samples were filtered for suspended solids with a 0.7-µm filter membrane (Whatman, GF/F)[57].

**Chemical analysis: POC and PN.** Particulate organic carbon (POC) and PN were determined from a selection of SS samples collected across the hydrograph (Fig. 1 and See Supplementary Fig. 3). Because the two catchments are comprised of igneous and volcaniclastic rocks, the river POC did not contain petrogenic organic carbon[9]; thus, all of the river organic carbon measured was considered biogenic.

Most of the WEBB[9] and LTER SS samples had been combusted at 550 °C to determine mass lost on ignition (LOI), which mainly removes all non-structural water trapped within clays and organic matter[9]. The resulting ashed weight essentially represents the bedrock contribution to the sediment load. Because of the high amounts of water loss in these samples compared to organic matter content, we determined that LOI was not a reliable measure for river particulate organic matter content.

Two methods were used to determine carbon content: elemental analysis and coulometry. A selection of 297 WEBB and LCZO SS samples were analysed by elemental analysis, which involved either grinding an SS sample or analysing the sample on the filter membrane (removing as much filter as possible)[57] and measuring POC% and PN% using a Costech CHNSO 4010 Elemental Analyzer at the University of Pennsylvania. Additionally, 96 WEBB SS samples were analysed for OC% using a Coulometrics Total Carbon Combustion Apparatus Model 5120 at the USGS Water Resources Mission Area Laboratory in Boulder, Colorado[55] (See Supplementary Data 1).

There are some small differences between the biogenic POC – SS concentration relations at SS concentrations <5 mg L$^{-1}$ (See Supplementary Data 1), which may be due to (1) protocol differences between the USGS WEBB and LCZO projects and/or (2) natural variation in the rivers at low flow, where POC% ranges from 1 to 13%[57]. The regions of variation are at low flows, however, and are not likely to influence annual river POC fluxes, which are driven by high flows. The samples, with slight variations, occur at low SS concentrations, which are lower than previously reported in the literature[8]. These petrogenic-carbon-poor rivers, which have lower SS concentrations than other drainages during similar runoff rates, represent a unique challenge in quantifying POC content at low flows. In this case, large volumes of water (e.g., 20 L) should be filtered at low flow in petrogenic-carbon-poor rivers to collect sufficient SS to properly quantify the POC%[57].

**Load and yield modeling**. The SS data were censored to obtain model results with acceptable biases using the following criteria: date and time of sample were known; measured instantaneous discharge was available; SS ≥3.5 mg, and ≥800 ml had been filtered from ~1000 ml. Additionally, any additional SS samples with measured POC and PN data were included even if they did not meet the selection criteria. This filtering process excluded 2.2% of samples from analysis (See Supplementary Table 1).

We estimated SS loads using Load ESTimator (LOADEST)[20,58]. LOADEST results are adjusted for log-transformed data. The LOADEST estimation file consisted of hourly discharge for the length of the study period without gaps, which was extracted at the top of the hour from the instantaneous river discharge datasets for both rivers[22]. The LOADEST calibration file contained the individual SS concentrations, with the date, time, and instantaneous discharge at collection. The calibration file consisted of SS samples from the USGS dataset from 1991 to 2015 and the LCZO/LTER dataset from 1997 to 2015 (See Supplementary Data 1). The USGS river SS datasets were previously used to determine suspended solid (SSol) yields from 1991 to 2005[59]. Because two datasets with different field sampling strategies and collection periods (See Supplementary Fig. 2) were included, LOADEST regression model #2 was used because it does not incorporate time-related changes in constituents in the model[20].

To extend the effective range along the hydrograph, POC and PN concentrations were estimated for SS samples without POC or PN measurements (Fig. 1; See Supplementary Data 1 and Supplementary Table 2). Constituent concentrations (mg L$^{-1}$) were log-log transformed and correlation coefficients regressions were generated. POC concentrations (mg L$^{-1}$) were estimated as:

$$POC_{est\ Icacos} = 0.054 \times SS^{0.977}, r^2 = 0.96 \qquad (5)$$

$$POC_{est\ Mameyes} = 0.069 \times SS^{0.977}, r^2 = 0.91 \qquad (6)$$

As a check on the estimated POC concentrations, we assessed the maximum possible POC concentration in the SS sample based on LOI, as thousands of SS samples were assessed for LOI. The organic matter contribution to LOI was calculated as twice POC. Whenever twice POC exceeded LOI or estimated LOI (see Stallard[22] which has the equations for estimated LOI using SS), POC was set to half the LOI. A handful of samples with estimated POC were capped and the maximum LOI estimated POC concentration was used instead. This is important because, at the highest discharges, all LOI (which was measured for most samples) appears to be POC. This implies that the bulk of the suspended load at high discharges is relatively low in clay minerals with loosely associated water. For the PN regressions see the Supplementary Information Table 2. We assessed POC and PN loads in LOADEST using the measured and estimated POC and PN sample concentrations.

Additionally, we estimated dissolved organic carbon (DOC) and dissolved inorganic carbon (DIC) loads using measurements from the WEBB program[9,22] (See Supplementary Data 1). River alkalinity, from bedrock-derived constituents accounting for atmospherically derived contributions, was estimated for the WEBB river samples and is equated with river DIC[9,22] (See Supplementary Data 1). DOC and DIC river yields were estimated in LOADEST, extending published yields by incorporating an additional 10 years of river discharge data (See Supplementary Data 2). Coarse POC[60,61] is likely an additional source of river POC during extreme storm events but was not quantified here.

The estimated daily loads (kg d$^{-1}$) outputs from LOADEST were summed to estimate annual yields, where all the constituents in our study had an $r^2 > 0.93$. The load bias for POC at Icacos and Mameyes was 8.5 and 28.3%, respectively, with positive numbers indicating a possible overestimation of daily loads, because we are modeling log-transformed data. While Mameyes may have an overestimation of daily loads (3.3% greater than the USGS recommended maximum), we consider the model to be at an acceptable level for this study[58] because of the exponential relationship that river suspended load has with discharge and an extremely wide range of stream discharges at this site.

**Other catchment carbon export estimates**. The burial efficiency of POC was conservatively estimated at 22 ± 5%, using the global terrestrial deltaic burial efficiency in marine deposits[28], for both the Icacos and Mameyes. Biogenic POC$_{burial}$ was estimated as:

$$POC_{burial}\ yield = biogenic\ POC\ yield \times POC\ burial\ efficiency$$

The carbon budget was determined by adding the DIC yield and the biogenic POC$_{burial}$ yield for each river. Because there is little to no petrogenic OC oxidation or sulfide oxidation, which are both effective sources of carbon[9], the two catchments are both carbon sinks and thus have substantial negative values.

We estimated the basin average total NPP based on the proportion of each forest type. In ArcMap 10.8.1 we extracted the landcover for the study catchments, using a 15-meter resolution landcover map generated from satellite imagery from 1999–2003 for Puerto Rico by Gould, et al.[62] and Gould, et al.[46]. Because NPP was measured in the Luquillo Mountains for the common vegetation types, we visually inspected the raster and assigned other vegetation types (young secondary wet montane forest, wet shrubland/grassland, and moist grassland/pastures) to one of the main vegetation types based on its proximity. These reallocated vegetation types represented 2.3 and 3.0% of the catchment area in Icacos and Mameyes, respectively. Given the simplifications outlined above, the Icacos catchment

(upstream of the stream gage) consists of Palo Colorado forest (*Cyrilla racemiflora*, 87.1%), and Sierra Palm forest (*Prestoea acuminata* var. *montana*, 12.9%). The Mameyes catchment (upstream of the stream gage) consists of Tabonuco forest (*Dacryodes excelsa*, 58.1%), Sierra Palm forest (26.0%), and Palo Colorado forest (14.7%), elfin woodlands/cloud forest (*Tabebuia rigida*–*Eugenia borinquensis*, 1.2%), and with a small proportion of riverbed and infrastructure (0.1%). The proportion of vegetation types reported here is similar to those reported for the study catchments in Murphy, et al.[15].

The total NPP consists of above-ground and below-ground NPP. There are several forest plots in the Luquillo Mountains where NPP was measured in the late 20th century[45]. The mean NPP of the forests types of the Luquillo mountains are: 949 ± 140 tC km$^{-2}$ yr$^{-1}$ (six measurements) for Tabonuco forest, a subtropical wet forest; 1220 tC km$^{-2}$ yr$^{-1}$ (one measurement) for Sierra Palm forest, a lower montane wet forest; 956 ± 1 tC km$^{-2}$ yr$^{-1}$ (two measurements) for Palo Colorado forest, a lower montane wet forest; and 342 ± 293 tC km$^{-2}$ yr$^{-1}$ (two measurements) for elfin forest, lower montane rain forest[45]. With the NPP data available, we estimate that the basin average NPP, based on the proportion of each forest type, is 990 ± 1 tC km$^{-2}$ yr$^{-1}$ for the Icacos catchment and 1013 ± 85 tC km$^{-2}$ yr$^{-1}$ for the Mameyes catchment. The estimated percentage of NPP$_{export}$ as river biogenic POC was calculated as:

$$NPP_{export}(\%) = POC_{bio}\ yield\ (t\,C\,km^{-2}\ yr^{-1})/Catchment\ NPP\ (t\,C\,km^{-2}\ yr^{-1}) \times 100$$

**Rainfall and extreme events**. Mean annual precipitation was obtained from Murphy, et al.[17]. Catchment-wide annual rainfall was determined for Mameyes and Icacos, using a combination of spatial interpolation with existing rainfall and a linear model of annual runoff versus annual precipitation[16] (See Supplementary Data 2).

Rain events for the USGS 50075000 Río Icacos rain gage were determined from 1993 to 2015, based on a 15-min rainfall record (5-min intervals during heavy rainfall). A new rain event was defined as a >3 h break in rainfall[63]. River fluxes were extracted based on three groups of rain events: <22 mm per event (including flow during dry periods), 22–92 mm per event, or >92 mm per event, with the cut-offs based on the 90th (22 mm per event) and the 98th (92 mm per event) percentiles. To determine stream discharge by rain event, a maximum buffer of 6 h on either side of an event was included to ensure storm-driven export from the Mameyes was incorporated because we used a rain gage in the Icacos to establish event timing in the Mameyes (no equivalent 15-min record is available in upper Mameyes). The buffer was cut short if it overlapped with another event or that event's buffer, and the shared time was divided between the two events. The daily river loads were broken down into hourly exports by dividing by 24 and the river fluxes were extracted for each event. Any discharges less than the 50th percentile were excluded from the calculation[9].

**Global river relationships**. Global compilations of river POC and SS yields were published by Hilton and West[2] and Galy, et al.[3] (See Supplementary Data 3). We separated these rivers into two groups based on petrogenic carbon content in the suspended load and defined petrogenic-carbon-poor sediment rivers as having a petrogenic POC yield <2% of the total POC yield. All petrogenic-carbon-rich bedrock is sedimentary. There were 16 rivers in the petrogenic-carbon-poor group, which includes tropical, temperate, and arctic regions[2,3,5,6,34,64–67] (See Supplementary Data 4). All rivers with a petrogenic POC yield >2% of the total POC yield we defined as petrogenic-carbon-rich, with moderate to high levels of petrogenic POC sediment. The majority (84 of 99) of global rivers are in the petrogenic-carbon-rich group[3,35]; many of the montane rivers with very high SS yields are in Taiwan and New Zealand[2] (See Supplementary Data 3).

The rivers included in the NPP$_{export}$ (%) relationships with SS yields were from Galy, et al.[3]. An additional publication[68] was included in the dataset for the NPP$_{export}$ (%) that had been published since the compilation article was published (See Supplementary Data 6). The results from our study were incorporated as well. There were 46 rivers with NPP$_{export}$ (%) included in the petrogenic-carbon-rich dataset with 50 entries (See Supplementary Data 6). There were nine rivers with NPP$_{export}$ (%) included in the petrogenic-carbon-poor dataset with 58 entries; 50 of which were from Icacos and Mameyes (See Supplementary Data 4).

We determined global river relationships between biogenic POC yield and SS yield and NPP$_{export}$ (%) and SS yield for petrogenic-carbon-poor and petrogenic-carbon-rich rivers. Because there are exponential relationships between biogenic POC yield and SS yield, and NPP$_{export}$ (%) and SS yield, a logarithmic transform was taken of each value, and then a linear regression was determined for each of the groups. These equations were then converted into exponential regression equations.

## Data availability

The river discharge dataset used in this paper are publicly available and can be downloaded from the US Geological Survey (USGS) surface-water historical instantaneous data for the Nation at https://doi.org/10.5066/F7P55KJN. All river constituent source data, from both the USGS and the Long-Term Ecological Research Network (LTER)/Luquillo Critical Zone Observatory (LCZO) are located in Supplementary Data 1. The storm event dataset was sourced from Scholl and Murphy[63].

## Code availability

The Load Estimator (LOADEST) software was developed by the USGS and can be downloaded from https://water.usgs.gov/software/loadest/index.html.

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

## Acknowledgements

We gratefully acknowledge support for K.E.C., A.F.P., G.G., and W.H.M. by the NSF Critical Zone Observatory Program (EAR-1331841 to A.F.P., G.G., and W.H.M.), and for R.F.S., S.F.M., and M.A.S. from the US Geological Survey Climate Research and Development program. The authors thank M.C. Larsen, A. Torres-Sánchez, M. Salgado, S. Moya, C. Estrada, M. Ittayem, J.F. Saraceno, D.R. Vann, M.L. Nunez and many others who carried out field sampling, sample processing, laboratory support, and analytical analyses, J.B. Shanley, J.K. Willenbring, D.J. Jerolmack for advice, and A. Ibáñez for translating the abstract into Spanish. Additional support was provided by the USDA Forest Service, the University of Puerto Rico, and the LUQ LTER program (NSF DEB-1546686 and NSF DEB-1831592 to G.G. and W.H.M.). This paper was greatly improved by suggestions by Kimberly Wickland (USGS) and reviewers Timothy Eglinton and Jin Wang. Any use of trade, firm, or product names is for descriptive purposes only and does not imply endorsement by the US Government. River suspended sediment samples were collected in accordance with relevant permits and local laws.

## Author contributions

Field collections were coordinated by R.F.S., G.G., and W.H.M. W.H.M., R.F.S., and A.F.P. conceived of the study. K.E.C., R.F.S., S.F.M., and M.A.S. contributed to the intellectual input of the data analysis. R.F.S. and K.E.C. carried out the sample and data analyses. K.E.C. drafted the manuscript with input from all authors.

## Competing interests

The authors declare no competing interests.
