## [Peer Review File · Nature Communications]

Title: Extreme rainstorms drive exceptional organic carbon export from forested humid-tropical rivers in Puerto RicoREVIEWER COMMENTS

Reviewer #1 (Remarks to the Author):

The authors present an extensive suite of concentration measurements on suspended sediment and particulate organic carbon (and nitrogen) for rivers draining 2 small catchments in eastern Puerto Rico that span a broad range of discharge levels and encompass numerous extreme rainfall events over a ~ 25-year time period. Sediment export from these heavily forested tropical watersheds developed on granitic bedrock is dominated by storm events. The authors find that POC-suspended sediment relationships for these watersheds differ from other, more commonly studied systems that are underlain by sedimentary rocks, reflecting contrasting erosional properties. Moreover, due to the absence of petrogenic (sedimentary rock-derived) organic carbon, the authors are able to convert carbon export to biospheric OC yields. They find that the higher biospheric POC export from these catchments is under-predicted from conventional POC-suspended sediment yield relationships, and represents a significant atmospheric CO₂ sink. The high percentage of net primary productivity (NPP%) export as POC for the Icacos river (as a consequence of relatively low NPP compared to POC export) is also interesting.

It is relatively rare to have such an extensive and extended record of POC and suspended sediment export from a catchment, in particular for these unrepresented systems. The observation of relatively low suspended sediment yields is interesting, and contrasts with many other tropical settings dominated by extreme events (e.g., Taiwan) upon which much of our current understanding is built. The overall contribution (2 to 4% of global terrestrial POC erosion) is quite modest and certainly within the uncertainty of global estimates, but nonetheless the authors make a strong case for separately considering and accounting for these systems in global estimates. I therefore think this is an important contribution that provides a further perspective on fluvial export of biospheric organic carbon. That said, while evidence for distinctive export of POC from the studied watersheds is strong, the link to burial is less clear. In this context, I have several comments and questions concerning the paper:

1. Uniqueness of systems investigated:

Emphasis is placed on the contrasting properties of volcanic mountain islands in the (Neo)tropics with rapidly uplifting (and eroding) mountain islands in the tropics (e.g., Taiwan). To what extent is the presence of these systems in the tropics critical relative to volcanic islands at higher latitudes (e.g., Iceland), or to younger tropical volcanic islands (e.g., Hawaii)?

2. POC vs DOC:

Although the current paper is focused on POC, in part due to its importance as a longer term carbon sink (through burial), is there any information on the balance of dissolved organic carbon (DOC) versus POC export/yields from these neotropical volcanic watersheds? In particular, given the contrasting NPP% export accounted for by POC, is this also the same for DOC or is the latter proportionally diminished? Furthermore, is there any information on the balance of POC and DOC export vary during extreme events in these catchments?

3. Nature of exported POC:

What is the form of the POC that is exported from these catchments? If, as implied from the differing POC-SS relationships, it is not strongly associated with mineral phases (due to resistance of older igneous rocks to erosion), then it would suggest that much is transported as mineral-free organic detritus (e.g., forest litter, upper organic layer of soils)? [Line 510: "... at highest discharges, all LOI [...] appears to be POC." This suggests POC is pure vegetation debris and minimal clays]. Is there any information on POC size (e.g., is it mostly coarse or fine-grained particles)? This brings up a key related question: What the degradability of this material is - i.e., to what extent does it survive transport, intermediate storage in any fluvial deposits and export? The current view of POC carried by large rivers is that much of it is (mineral) soil derived and the degradability of this material may differ (lower?) than that in the current systems. This, in turn, may affect the balance between export and remineralisation of POC entering the rivers. In general, the paper would benefit from more in-depth discussion of the mechanistic context/explanation for the observations.

4. POC burial:

Export of POC is one thing, burial is another. It is stated (Line 181-183) that "These extreme events are responsible for approximately half of the annual biospheric POC burial in the adjacent ocean (if burial rate of POC is conservative and constant; Figure 3; Table S6)." The latter are important assumptions. To what extent can they be supported? In the Methods section, it is mentioned that burial efficiencies of 30% and 40% are estimated from the Hilton & West papers.

These burial efficiencies are based on rivers with predominantly high SS concentrations (i.e., high mineral-associated POC proportions) with this mineral association enhancing the burial potential of exported POC. These burial efficiencies may be too high given the relatively low suspended sediment yields of the two catchments. What is meant by POC burial being conservative, particularly in the context of its differing mode or extent of mineral association? Since extreme events are responsible for most POC export, and during such events export should be rapid and bypass intermediate storage reservoirs (e.g., floodplains), but its fate (burial versus remineralisation) upon discharge and dispersal in the ocean it less clear. Where is this burial inferred to take place? What is fate of exported water, sediment and POC during extreme events (hyperpycnal flows)? Overall, more rigorous discussion and justification of the burial estimates would be helpful.

5. Relevance to global estimates:

It is mentioned that global terrestrial biosphere erosion may be underestimated by 2.1 to 4.4 %. There is a lot of uncertainty in current estimates due to the intrinsic scatter in available data and in [logarithmic] relationships between SS and POC yields (Galy et al 2015 paper). To what extent is this underestimation significant given these inherent uncertainties?

Other comments/questions:

- It would help to have more information concerning the nature of the catchments, and the specific

sampling locations.

- Concluding paragraph (line 197-198): "The most important factors influencing river biospheric POC export in Luquillo are: (1) the catchments drain rocks without petrogenic carbon (or reduced sulfur), ...". It is not clear to me how biospheric POC export is influenced by petrogenic carbon. Is it not that it is the association yield of fine-grained minerals from these readily erodible rocks that is the critical factor?

- It is mentioned (line 573-574) that "all petrogenic C rich bedrock is sedimentary". Therefore isn't it the presence/absence of the sedimentary rock, rather than the petrogenic C, that is key?

- The separate regressions that are proposed - should these be applied to all petrogenic C poor rivers, or only those in neotropical regions?

- Line 167–171. Why are erosion rates high, but SS yields only moderate in the Icacos catchment?

Reviewer #2 (Remarks to the Author):

In this paper, Clark and co-workers present an interesting study on the role of extreme rain events on the carbon export in a tropical igneous area. To do this, they used some amazing long-time data – the 25-year records of carbon and sediment. They found the two Puerto Rico rivers have very large biospheric POC yield. Extreme rainfall events play a central role on the carbon export. Counting in silicate weathering, these two rivers have high rate of atmospheric CO₂ sink compared to other regions. They found that the biospheric POC flux of such wet tropical area has been underestimated on the global river POC export model. They suggested that wet tropical mountain islands may contribute an additional 2-4% of the biospheric POC erosion, which is a geological carbon sink.

This paper is generally well-written. An assessment of the role of extreme rainfall on the carbon cycle is important as extreme rain events are reported to be more frequent in the future. This paper would be interested a broad earth science community that mean it could be well suited to the readers of nature communications.

While the dataset is unique, and the result is a valuable contribution to knowledge of river organic carbon cycle, I have some major concerns below, followed by a list of detailed review and comments which need to be addressed before publishing.

1. The way of estimating the biospheric POC for the whole tropical volcanic mountains: The assumption of "these islands have biospheric POC yields like the Mameyes" is not convincing, which make the conclusion of "underestimated by 2% to 4%" is too weak. I think that you could use Galy et al. (2015) method but with your petrogenic-carbon-poor aggression, and the mean suspended sediment yield of the tropical volcanic catchment.

2. I found that POC%, SS concentration are significantly different between 1991-2004 and 2005-2015 (attached figure). Is it because of sampling bias? If yes, a clarify of whether it bias the flux calculation

may be needed. If no (I guess this is true), is there another story behind there? It this because of land-use change, or climate? The long-term record of runoff and sediment concentration may be very helpful to assess what had happen during the study period. I think that the analysis of the temporal variation of sediment and biospheric POC and their controlling factors would be very interesting.

Furthermore, Are the sediment samples still available? If yes, the radiocarbon analysis of the POC will help to investigate the organic carbon source and turnover time, which will much strengthen the manuscript.

Minor comments:

Title: perhaps replace “extreme events” by “extreme rain event”.

Line 38: may add “mainly” in front of “depending” – geology is not the only controlling factor on the river carbon budget.

Line 48: I think it is better to specify that the silicate weathering is another important geological carbon sink in this paragraph. The study also mentioned that the tropical igneous river is important for the carbon cycle because of the low content of pyrite and petrogenic organic carbon. Therefore, I suggest that some sentences could be added to generally introduce other geological carbon processes.

Line 66: I would suggest specifying the river is “mountain river” since the passive rivers also flow directly into the ocean but their biospheric POC burial efficiency is lower.

Line 79: consider giving the detail number of catchment area here.

Line 82: the mean annual rainfall value needs a reference here or in Table 1.

Line 83: consider giving the timespan here – “over 25 years from 1991 to 2015”.

Line 85: numbers of the fluxes come out a little suddenly. I would suggest adding one or two sentences to introduce how the fluxes are calculated here.

Line 87: please check the format of the reference citation of Nature Communication. I guess a bracket is needed for the cited reference number when it is behind a superscript number.

Line 88-92: it is unclear why the study of ref. 15 is also underestimated. I think that using the loss on ignition of sediment as the POC%, it will be overestimated. Please clarify it.

Line 114: it is worth summarizing how you calculated the biospheric POC burial flux. In addition, I was wondering if using 40% and 30% as the POC burial efficiency is appropriate. I would suggest some discussion on your choice. In my opinion, this may be a conservative estimate. The extreme rain event

dominates the POC export, and the turbidity generated during the event could increase the POC preservation during the deposition. Anyway, the discussion on the POC burial efficiency and its uncertainty will strengthen the conclusion of this study.

Line 122: maybe add a sentence says, the left carbon sink is attributed by the silicate weathering which generate DIC.

Line 122: The largest measured carbon sink is in term of yield, not flux. Please clarify it. Same in Line 130, the studied rivers are small, thus it may not suit to say they are important carbon sinks.

Line 135: Please report propagated measurement uncertainty for the POC yields.

Line 139-140: The bedrock has no biospheric carbon, so I think that the carbon content of bedrock cannot influence the biospheric POC yields.

Line 144. How is the value 0.64 calculated? Please clarify it as it is important for assessing the importance role of tropical volcanic mountains on the carbon cycle.

Line 151: it is cool that you find the distinct relationships and the cut-off criterion. However, I found that concept between “bedrock petrogenic OC content” with “riverine petrogenic POC content” is mixed up. I believe that low bedrock petrogenic OC content is always accompanied with low riverine petrogenic POC content. But they are still not equal. Please clarify it.

Line 183: consider re-phrasing “while counting in the carbon sink from the silicate weathering, 44% of the carbon” .

Line 431: a little more detailed summary on the precipitation, erosion, turnover and chemical weathering is needed here.

Line 462: please define the abbreviation LCZO here.

Line 459: please add the information of the sampling frequency and method.

Line 508-514: consider re-phrasing “The POC contribution to LOI was calculated as 50%”. In addition, what your aim for this paragraph? It is not clear.

Figure 1: it seems that the samples with the largest suspended sediment concentration are not included in the POC% analysis. Will it influence the flux calculation?

Figure 2: please use a different symbol for the River Icacos and Mameyes.

Figure S2: the symbol is not very clear. Please consider using colour symbols.

Jin Wang,
Institute of Earth Environment, Chinese Academy of Science.

Response to Editor and Reviewers' Comments for Nature Communications manuscript ID: NCOMMS-21-27380-T

Comments are copied below with our responses in green text, manuscript text is in quotes with small additions to quoted text being bolded.

Reviewer #1 (Remarks to the Author):

The authors present an extensive suite of concentration measurements on suspended sediment and particulate organic carbon (and nitrogen) for rivers draining 2 small catchments in eastern Puerto Rico that span a broad range of discharge levels and encompass numerous extreme rainfall events over a ~ 25-year time period. Sediment export from these heavily forested tropical watersheds developed on granitic bedrock is dominated by storm events. The authors find that POC-suspended sediment relationships for these watersheds differ from other, more commonly studied systems that are underlain by sedimentary rocks, reflecting contrasting erosional properties. Moreover, due to the absence of petrogenic (sedimentary rock-derived) organic carbon, the authors are able to convert carbon export to biospheric OC yields. They find that the higher biospheric POC export from these catchments is under-predicted from conventional POC-suspended sediment yield relationships, and represents a significant atmospheric CO₂ sink. The high percentage of net primary productivity (NPP%) export as POC for the Icacos river (as a consequence of relatively low NPP compared to POC export) is also interesting.

It is relatively rare to have such an extensive and extended record of POC and suspended sediment export from a catchment, in particular for these unrepresented systems. The observation of relatively low suspended sediment yields is interesting, and contrasts with many other tropical settings dominated by extreme events (e.g., Taiwan) upon which much of our current understanding is built. The overall contribution (2 to 4% of global terrestrial POC erosion) is quite modest and certainly within the uncertainty of global estimates, but nonetheless the authors make a strong case for separately considering and accounting for these systems in global estimates. I therefore think this is an important contribution that provides a further perspective on fluvial export of biospheric organic carbon. That said, while evidence for distinctive export of POC from the studied watersheds is strong, the link to burial is less clear. In this context, I have several comments and questions concerning the paper:

We would like to thank the reviewer for their feedback. We have revised the abstract to reflect your comments. We have re-evaluated how we assess the underestimation of OC export in the geological carbon cycle and explained why this potential underestimate is important. We have revised our POC burial estimates using more conservative burial rates.

We have modified the description of our data in the **methods** to emphasize the uniqueness of this study. Specifically, we collected far more samples of most dissolved and suspended constituents than in other studies, notably those from Taiwan and New Zealand. As such, we have added the following text to lines 582 to 590 in the revised version:

“During the USGS WEBB program, far more samples of most dissolved and suspended constituents were collected than in other studies. Most importantly, SS samples analyzed for biogenic POC were collected during extreme rainfall events where runoff was $> 10 \text{ mm hr}^{-1}$ at both Icacos ($n = 1$) and Mameyes ($n = 7$); the number of samples analyzed for SS (used to estimate POC and PN, see below) and comprehensive chemistry was considerably greater (See Supplementary Table 1). There have only been two other rivers where SS samples analysed for biogenic POC were collected at extremely high runoff, specifically 11.2 mm hr^{-1} in Taiwan¹ and 11.1 mm hr^{-1} in the Erlenbach in the Swiss Alps².”

1. Uniqueness of systems investigated:

Emphasis is placed on the contrasting properties of volcanic mountain islands in the (Neo)tropics with rapidly uplifting (and eroding) mountain islands in the tropics (e.g., Taiwan). To what extent is the presence of these systems in the tropics critical relative to volcanic islands at higher latitudes (e.g., Iceland), or to younger tropical volcanic islands (e.g., Hawaii)?

We have revised the global petrogenic-carbon-poor river dataset and it now includes three Hawaiian Islands. The biogenic POC – SS yield Equation 1 now reflects these rivers. In the text we have added the following reference to Hawaii on line 44:

“Igneous montane islands, such as in Guadeloupe^{3,4} and Hawaii⁵, are distinct because these largely contain only carbon sinks.”

In the revised version we have put less emphasis on tropical volcanic islands. Icelandic river POC yield datasets for solely igneous bedrock were not available to use in our regression. Other northern rivers that were petrogenic-carbon-poor, such as the St. Lawrence River, was included in the regression.

2. POC vs DOC:

Although the current paper is focused on POC, in part due to its importance as a longer term carbon sink (through burial), is there any information on the balance of dissolved organic carbon (DOC) versus POC export/yields from these neotropical volcanic watersheds? In particular, given the contrasting NPP% export accounted for by POC, is this also the same for DOC or is the latter proportionally diminished? Furthermore, is there any information on the balance of POC and DOC export vary during extreme events in these catchments?

Based on your suggestion we have added the following text on lines 217 to 225:

“Annual biogenic POC and DOC yields are both higher in the Icacos than the Mameyes. The biogenic POC:DOC for the Mameyes, 2.0 ± 0.2 , is similar to that of the Capesterre catchment in Guadeloupe at $2.5^{3,4}$, whereas the Icacos has a ratio of 5.0 ± 1.0 , indicating dominance of particulates in river OC export (See Supplementary Table 2). The ratio increases during extreme rainfall events and decreases during small rainfall events ($< 22 \text{ mm per event}$) and dry periods (See Supplementary Table 6). Almost twice as much $\text{NPP}_{\text{export}}$ (%) as DOC ($\text{NPP}_{\text{export_DOC}}$) is exported from the Icacos than from the Mameyes, whereas in the case of $\text{NPP}_{\text{export}}$ (%) as POC the Icacos is more than three times larger than from the Mameyes (Table 1; See Supplementary Table 2).”

3. Nature of exported POC:

What is the form of the POC that is exported from these catchments? If, as implied from the differing POC-SS relationships, it is not strongly associated with mineral phases (due to resistance of older igneous rocks to erosion), then it would suggest that much is transported as mineral-free organic detritus (e.g., forest litter, upper organic layer of soils)? [Line 510: "... at highest discharges, all LOI [...] appears to be POC." This suggests POC is pure vegetation debris and minimal clays]. Is there any information on POC size (e.g., is it mostly coarse or fine-grained particles)? This brings up a key related question: What the degradability of this material is - i.e., to what extent does it survive transport, intermediate storage in any fluvial deposits and export? The current view of POC carried by large rivers is that much of it is (mineral) soil derived and the degradability of this material may differ (lower?) than that in the current systems. This, in turn, may affect the balance between export and remineralisation of POC entering the rivers. In general, the paper would benefit from more in-depth discussion of the mechanistic context/explanation for the observations.

To provide additional information, the following paragraph was inserted into the methods section on lines 654 to 664:

"As a check on the POC estimated concentrations, we assessed the maximum possible POC concentration in the SS sample based on LOI, as thousands of SS samples were assessed for LOI. The organic matter contribution to LOI was calculated as twice POC. Whenever twice POC exceeded LOI or estimated LOI (see Stallard⁶ which has the equations for estimated LOI using SS), POC was set to half the LOI. A handful of samples with estimated POC were so capped and the maximum LOI estimated POC concentration was used instead. This is important, because at highest discharges, all LOI (which was measured for most samples) appears to be POC. This implies that the bulk of suspended load at high discharges is relatively low in clay minerals with loosely associated water. For the PN regressions see the Supplementary Information Table 8. We assessed POC and PN loads in LOADEST using the measured and estimated POC and PN sample concentrations."

While you raise some interesting questions about POC degradability, there is little information in the literature of Puerto Rico for this question. We have revised our POC burial yield estimates, using a more conservative value for global rivers. We have added the following text on lines 111 to 125:

"In the context of the geological/long-term carbon cycle, however, the burial of the exported biogenic POC is most relevant. Although there have not been any direct measurements of biogenic POC_{burial} in the coastal regions of Puerto Rico, rivers discharge onto the shallow submarine shelf, where river POC spreads out and is reworked in the oxygenated environment⁷. It is expected that Icacos and Mameyes have a lower burial efficiency because they have lower SS yields⁸. During extreme events, when over half of biogenic POC export occurs (See Supplementary Table 6), the POC is mainly derived from organic soil and vegetation, but may be converted back to carbon dioxide in the marine environment before deposition (approximately 90% of organic matter in marine deposits is associated with minerals, such as clays and silts⁹). Without measurements available from the ocean adjacent to eastern Puerto Rico, we estimate

biogenic $\text{POC}_{\text{burial}}$ using the global river estimates of $22 \pm 5\%$ ¹⁰. The biogenic $\text{POC}_{\text{burial}}$ is estimated at $14.3_{-6.0}^{+7.7}$ and $3.9_{-1.7}^{+2.3}$ $\text{tC km}^{-2} \text{ yr}^{-1}$ for the Icacos and Mameyes, respectively (Table 1; See Supplementary Table 2). The Icacos biogenic $\text{POC}_{\text{burial}}$ yields overlap with the Hokitika and the Whatoroa rivers, in New Zealand, and yields from the Mameyes overlap with the Peel River and the Arctic Red, in Canada, and the Capesterre, in Guadeloupe⁸.”

4. POC burial:

Export of POC is one thing, burial is another. It is stated (Line 181-183) that “These extreme events are responsible for approximately half of the annual biospheric POC burial in the adjacent ocean (if burial rate of POC is conservative and constant; Figure 3; Table S6).” The latter are important assumptions. To what extent can they be supported? In the Methods section, it is mentioned that burial efficiencies of 30% and 40% are estimated from the Hilton & West papers. These burial efficiencies are based on rivers with predominantly high SS concentrations (i.e., high mineral-associated POC proportions) with this mineral association enhancing the burial potential of exported POC. These burial efficiencies may be too high given the relatively low suspended sediment yields of the two catchments. What is meant by POC burial being conservative, particularly in the context of its differing mode or extent of mineral association? Since extreme events are responsible for most POC export, and during such events export should be rapid and bypass intermediate storage reservoirs (e.g., floodplains), but its fate (burial versus remineralisation) upon discharge and dispersal in the ocean it less clear. Where is this burial inferred to take place? What is fate of exported water, sediment and POC during extreme events (hyperpycnal flows?)? Overall, more rigorous discussion and justification of the burial estimates would be helpful.

We have thought about $\text{POC}_{\text{burial}}$ and how much we can really say about this topic without measurements. We have removed the text on $\text{POC}_{\text{burial}}$ during extreme events. Also, we have rethought the POC burial efficiency and decided to go with the global average for deltaic river sediments ($22 \pm 5\%$) (Burdige, 2005) rather than using the assumed burial efficiency (30% for Guadeloupe) from Hilton and West (2020). Thus, we have modified the text on burial on lines 120 to 121:

“Without measurements available from the ocean adjacent to eastern Puerto Rico, we estimate biogenic $\text{POC}_{\text{burial}}$ using the global river estimates of $22 \pm 5\%$ ¹⁰.”

When discussing the carbon budget, we have modified the text to focus on overall sinks, without a storm focus. This topic is very interesting however, and physical measurements of $\text{POC}_{\text{burial}}$ during extreme events would be an important component to understanding these mountains as carbon sinks. Thus, we have added the following text to lines 133 to 140:

“The carbon budgets in the Mameyes and Capesterre are dominated by chemical weathering of silicates (See Supplementary Table 2). This is not true for the Icacos, where physical erosion and subsequent $\text{POC}_{\text{burial}}$ may contribute over half of the carbon budget (See Supplementary Table 2), despite the fact that it has one of the fastest chemical weathering rates of granitic rocks in the world¹¹. To determine the true extent of which the Luquillo catchments are geological carbon sinks, a comprehensive assessment of river biogenic POC burial efficiency in the marine environment is needed.”

5. Relevance to global estimates:

It is mentioned that global terrestrial biosphere erosion may be underestimated by 2.1 to 4.4 %. There is a lot of uncertainty in current estimates due to the intrinsic scatter in available data and in [logarithmic] relationships between SS and POC yields (Galy et al 2015 paper). To what extent is this underestimation significant given these inherent uncertainties?

We thank you for your comment and you bring up a good point. We have revised the method for calculating the underestimation of global OC export following the suggestion by reviewer two. The upper range of our underestimation is higher now. We have revised the text found on lines 169 to 182:

“More broadly speaking, biogenic POC export from rivers draining humid-tropical igneous mountainous islands and island arcs, which encompass a total area of 320,000 square km^{4,12,13}, are also likely underestimated. By assuming that these rivers are similar to those of our study rivers, we estimated biogenic POC fluxes, using the SS yields from Mameyes as the lower bound and from Icacos as the higher bound. We determined the biogenic POC yields using the regression developed by Galy, et al.¹⁴ and then multiplied the area^{4,12,13} to approximate biogenic POC fluxes for this region. Based on the Galy, et al.¹⁴ regression, humid-tropical igneous montane rivers contribute 0.64 to 1.57 MtC yr⁻¹ of biogenic POC fluxes. On the other hand, following the same steps outlined above, but using Eq. 1, this region would contribute fluxes of 5.59 to 23.52 MtC yr⁻¹. This represents a potentially substantial underestimation of biogenic OC export from humid-tropical igneous mountainous islands in the geological carbon cycle^{8,14}. Accordingly, global biogenic OC export, which is currently estimated at 110-230 MtC per year⁸, may be underestimated by 2.1 to 8.7% yr⁻¹. As more research is conducted in these regions, their relative importance to global biogenic POC export will become more apparent.”

Other comments/questions:

- It would help to have more information concerning the nature of the catchments, and the specific sampling locations.

In the main text we have added more background for the rivers in the section River particulate organic carbon in the Luquillo Mountains on lines 80 to 85:

“The two study rivers, Icacos and Mameyes, are in the Luquillo Mountains in Puerto Rico (See Supplementary Fig. 1), and are both small catchments (3.26 km² and 17.8 km², respectively) with granodioritic (Icacos) and volcanoclastic (Mameyes) bedrock that do not contain substantial petrogenic organic carbon, reduced sulfur, or carbonates^{15,16}. These catchments have very high rainfall (average of 5,000 and 3,720 mm yr⁻¹ in the Icacos and Mameyes, respectively)^{17,18} and runoff (3,906 and 2,770 mm yr⁻¹, respectively) (Table 1).”

In the Methods section under study area, we have added text from lines 563 to 569:

“The sources of suspended sediment erosion from the study catchments are from surficial erosion, bed and bank erosion, and from landslides¹⁹. The total hillslope erosion is estimated to

be 750 and 523-2143 t km⁻² yr⁻¹ at the Icacos and Mameyes respectively, where the Mameyes had a particularly high prehistoric landslide erosion (2000 t km⁻² yr⁻¹)¹⁹. Chemical weathering rates of silicate rocks are among the fastest in the world in the Icacos¹¹, with a weathering front of approximately 1 cm per 100 years²⁰. The volcanoclastic rock of the Mameyes weathers event faster, and both catchments have a thick layer of saprolite^{11,21}.”

- Concluding paragraph (line 197-198): "The most important factors influencing river biospheric POC export in Luquillo are: (1) the catchments drain rocks without petrogenic carbon (or reduced sulfur), ...". It is not clear to me how biospheric POC export is influenced by petrogenic carbon. Is it not that it is the association yield of fine-grained minerals from these readily erodible rocks that is the critical factor?

You bring up a good point, and while that is a very interesting question, it is unclear if petrogenic POC interferes with biogenic POC transport in rivers. We have reworked this phrase now on lines 237 to 242:

“The most important factors influencing river biogenic POC export from Luquillo are: (1) igneous intrusive and volcanoclastic bedrock types, which are less erodible than sedimentary bedrock, (2) large organic carbon stocks available to be eroded from the landscape, and (3) high precipitation with frequent extreme rainfall events. These factors together are uncommon in previous studies of river biogenic POC export from montane regions.”

- It is mentioned (line 573-574) that “all petrogenic C rich bedrock is sedimentary”. Therefore isn't it the presence/absence of the sedimentary rock, rather than the petrogenic C, that is key?

This text is now on lines 741 to 742. Basically, all petrogenic-carbon-rich rocks are sedimentary, but all sedimentary rocks are not petrogenic-carbon-rich (such as most limestones, dolomites, and many sandstones and coarse-grained clastic rocks).

- The separate regressions that are proposed - should these be applied to all petrogenic C poor rivers, or only those in neotropical regions?

The proposed regression was created using data from global rivers that were petrogenic-carbon poor rivers (i.e. the St. Lawrence River). A large focus of our paper, however, is the humid-tropics and other high rainfall montane environments where distinct differences appear in the relationships between biogenic POC and SS yields in petrogenic-carbon-poor and petrogenic-carbon-rich rivers. Please see the text now on lines 742 to 743:

“There were **16** rivers in the petrogenic-carbon-poor group, which includes tropical, temperate, and arctic regions^{4,5,8,14,22-26} (See Supplementary Table 5).”

- Line 167–171. Why are erosion rates high, but SS yields only moderate in the Icacos catchment?

We modified this sentence now on lines 200 to 204:

“Although erosion rates are very high in the Icacos catchment **relative to other watersheds in eastern Puerto Rico**^{15,19}, SS yields are only considered moderate when compared to global montane rivers with sedimentary bedrock⁸; however, the Icacos has one of the highest estimated percentage of NPP_{export} by POC in the world, suggesting that separate relationships exist between these two bedrock types (Figure 2b).”

Reviewer #2 (Remarks to the Author):

In this paper, Clark and co-workers present an interesting study on the role of extreme rain events on the carbon export in a tropical igneous area. To do this, they used some amazing long-time data – the 25-year records of carbon and sediment. They found the two Puerto Rico rivers have very large biospheric POC yield. Extreme rainfall events play a central role on the carbon export. Counting in silicate weathering, these two rivers have high rate of atmospheric CO₂ sink compared to other regions. They found that the biospheric POC flux of such wet tropical area has been underestimated on the global river POC export model. They suggested that wet tropical mountain islands may contribute an additional 2-4% of the biospheric POC erosion, which is a geological carbon sink.

This paper is generally well-written. An assessment of the role of extreme rainfall on the carbon cycle is important as extreme rain events are reported to be more frequent in the future. This paper would be interested a broad earth science community that mean it could be well suited to the readers of nature communications.

While the dataset is unique, and the result is a valuable contribution to knowledge of river organic carbon cycle, I have some major concerns below, followed by a list of detailed review and comments which need to be addressed before publishing.

Thank you for your feedback. Please see our comments below.

1. The way of estimating the biospheric POC for the whole tropical volcanic mountains: The assumption of “these islands have biospheric POC yields like the Mameyes” is not convincing, which make the conclusion of “underestimated by 2% to 4%” is too weak. I think that you could use Galy et al. (2015) method but with your petrogenic-carbon-poor aggression, and the mean suspended sediment yield of the tropical volcanic catchment.

Thank you for this advice. We have changed the estimations in the manuscript using the mean SS yields of Mameyes (lower range) and Icacos (upper range) to represent global tropical-humid mountainous island and island arc catchments. We have used the method you suggested applying the SS yields for Icacos and Mameyes to the two regressions (Galy et al, 2015 and then ours with Eq. 1). The range is now 2.1 to 8.7%. We have moved this paragraph to after the introduction of Eq. 1. The text is now on lines 169 to 182:

“More broadly speaking, biogenic POC export from rivers draining humid-tropical igneous mountainous islands and island arcs, which encompass a total area of 320,000 square km^{4,12,13}, are also likely underestimated. By assuming that these rivers are similar to those of our study rivers, we estimated biogenic POC fluxes, using the SS yields from Mameyes as the lower bound and from Icacos as the higher bound. We determined the biogenic POC yields using the regression developed by Galy, et al.¹⁴ and then multiplied the area^{4,12,13} to approximate biogenic POC fluxes for this physiographic setting. Based on the Galy, et al.¹⁴ regression, humid-tropical igneous montane rivers contribute 0.64 to 1.57 MtC yr⁻¹ of biogenic POC fluxes. On the other hand, following the same steps outlined above, but using Eq. 1, this setting would contribute fluxes of 5.59 to 23.52 MtC yr⁻¹. This represents a potentially substantial underestimation of biogenic OC export from humid-tropical igneous mountainous islands in the geological carbon cycle^{8,14}. Accordingly, global biogenic OC export, which is currently estimated at 110-230 MtC

per year⁸, may be underestimated by 2.1 to 8.7% yr⁻¹. As more research is conducted in these regions, their relative importance to global biogenic POC export will become more apparent.”

2. I found that POC%, SS concentration are significantly different between 1991-2004 and 2005-2015 (attached figure). Is it because of sampling bias? If yes, a clarify of whether it bias the flux calculation may be needed. If no (I guess this is true), is there another story behind there? It this because of land-use change, or climate? The long-term record of runoff and sediment concentration may be very helpful to assess what had happen during the study period. I think that the analysis of the temporal variation of sediment and biospheric POC and their controlling factors would be very interesting.

Thank you for highlighting this. There were two different sampling campaigns that took place, with slightly different methods and collection regimes. The end of the USGS WEBB storm sampling occurred at the end of 2004, which is where the odd-looking pattern in SSC vs Q is coming from (See Supplementary Figure 2). We have clarified the storm sampling period in the text on lines 576 to 576:

“River samples were collected from 1991 to 2015, by the USGS WEBB program^{27,28}, **with a focus on sampling during large rainfall events from 1992 to 2004** (See Supplementary Fig. 2 and Table 1).”

In terms of the POC vs SS concentrations relationship inconsistencies that you point out, the storm sampling focus explains the lack of very high POC-SSC samples after 2004. The distinction at the lower end, may be due to differences between the two sampling campaigns, which we have previously outlined in the methods. Additionally, there is a natural variation in POC% at very low flows, that doesn't occur during higher flows. To address this issue, we have added the following text to the methods on lines 618 to 628:

“There are some small differences between the biogenic POC – SS concentration relations, at SS concentrations < 5 mg L⁻¹ (See Supplementary Table 1), which may be due to 1) protocol differences between the USGS WEBB and LCZO projects, and/or 2) natural variation in the rivers at low flow, where POC% ranges from 1 to 13%²⁹. The regions of variation are at low flows, however, and are not likely to influence annual river POC fluxes, which are driven by high flows. The samples, with slight variations, occur at low SS concentrations, which are lower than previously reported in the literature³⁰. Highlighting the distinctiveness of petrogenic-carbon-poor rivers, which have lower SS concentrations at similar discharges, and represents a unique challenge in quantifying POC content at low flows. In this case large volumes water (i.e. 20 L) should be filtered at low flow in petrogenic-carbon-poor rivers to collect sufficient SS to properly quantify the POC%²⁹.”

Furthermore, Are the sediment samples still available? If yes, the radiocarbon analysis of the POC will help to investigate the organic carbon source and turnover time,, which will much strengthen the manuscript.

Thank you for this suggestion and while those questions are out of the scope of the current paper, they are very interesting questions and potential next steps. This is a great idea for another paper investigating the landscape OC turnover time, especially as there is no petrogenic carbon (radiocarbon dead) in these catchments. In terms of identifying sources of OC within the river POC, radiocarbon could be helpful, but the vegetation and organic soil carbon signal may overlap in the mixing model. Another potentially interesting method would be to evaluate the leaf wax, based on long-chained *n*-alkanes (average chain length (ACL) and carbon preference index (CPI)) as this would allow for a distinction between soil and vegetation endmembers.

Minor comments:

Title: perhaps replace “extreme events” by “extreme rain event”.

We have changed the title to “Extreme rainstorms”

Line 38: may add “mainly” in front of “depending” – geology is not the only controlling factor on the river carbon budget.

This portion of the sentence was removed during revisions.

Line 48: I think it is better to specify that the silicate weathering is another important geological carbon sink in this paragraph. The study also mentioned that the tropical igneous river is important for the carbon cycle because of the low content of pyrite and petrogenic organic carbon. Therefore, I suggest that some sentences could be added to generally introduce other geological carbon processes.

The paragraph was modified from lines 44 to 52:

“Igneous montane islands, such as in Guadeloupe^{3,4} and Hawaii⁵, are anomalies because these largely contain only carbon sinks. There is a drawdown of atmospheric CO₂ when river particulate organic carbon (POC) from the biosphere is buried in long-term ocean deposits, and dissolved inorganic carbon (DIC) is produced during silicate weathering⁸. There is an effective release of CO₂ from rocks to the atmosphere when petrogenic organic carbon and sulfides in bedrock oxidize^{8,31}. Carbon budgets of humid-tropical igneous montane islands are rarely assessed, and yet the erosion of biospheric POC from these catchments play a potentially significant, yet poorly characterized, role in the geological carbon cycle.”

Line 66: I would suggest specifying the river is “mountain river” since the passive rivers also flow directly into the ocean but their biospheric POC burial efficiency is lower.

We have changed mountain river to montane river or mountainous river throughout the text.

Line 79: consider giving the detail number of catchment area here.

Done, now on line 81.

Line 82: the mean annual rainfall value needs a reference here or in Table 1.

We have now added a reference in the text and in the Supplementary Table 2. (now on line 84 to 85).

Line 83: consider giving the timespan here – “over 25 years from 1991 to 2015”.

Done (now line 90).

Line 85: numbers of the fluxes come out a little suddenly. I would suggest adding one or two sentences to introduce how the fluxes are calculated here.

We have ended the previous paragraph with a summary of the methods, so the yield results are less abrupt, now on lines 88 to 90:

“Using continuous river discharge measured by the U.S. Geological Survey (USGS)^{32,33}, concentration data (See Supplementary Table 1), and the USGS program LOADEST³⁴, we determined river yields from 1991 to 2015.”

Line 87: please check the format of the reference citation of Nature Communication. I guess a bracket is needed for the cited reference number when it is behind a superscript number.

We have changed these yields to read xx tC km⁻² per year to avoid confusion.

Line 88-92: it is unclear why the study of ref. 15 is also underestimated. I think that using the loss on ignition of sediment as the POC%, it will be overestimated. Please clarify it.

That sentence has been modified, as reference 15 (Stallard, 1988) had nothing to do with LOI measures. We realize how this explanation was confusing. Because of the clay water loss, LOI could not be used as the POC%. Only a portion of LOI was assumed to be POC, such that the average POC% of SS would be ~1% POC of SS. But we now know that the POC% of SS is actually much higher for these rivers. We have updated the text, now on lines 94 to 102:

“Previously reported river POC yields were underestimated because one study did not collect samples at the highest discharges³⁵ and in the other study^{6,15} POC% was estimated from a relation with “loss on ignition” (LOI) of sediment samples [mass lost when baked at high temperature (550°C), which includes organic matter plus water of hydration from clays¹⁵] and an average sediment/POC relation for global rivers³⁶ (See Supplementary Table 3) which is too low for these highly weathered, clay-rich catchments. In contrast to the previous work, this current study includes samples collected across a very wide range of runoff rates, and SS samples were analyzed for POC% with either an elemental analyzer or a carbon coulometer, rather than estimation from LOI.”

Line 114: it is worth summarizing how you calculated the biospheric POC burial flux. In addition, I was wondering if using 40% and 30% as the POC burial efficiency is appropriate. I would suggest some discussion on your choice. In my opinion, this may be a conservative estimate. The extreme rain event dominates the POC export, and the turbidity generated during the event could increase the POC preservation during the deposition. Anyway, the discussion on the POC burial efficiency and its uncertainty will strengthen the conclusion of this study.

We have reworked this text, and we have moved away from discussing $\text{POC}_{\text{burial}}$ specifically during storm events and focus more on POC export. The topic is very interesting though. We have revised the burial efficiency, which now is more conservative. The new text is from lines 111 to 125:

“In the context of the geological/long-term carbon cycle, however, the burial of the exported biogenic POC is most relevant. Although there have not been any direct measurements of biogenic $\text{POC}_{\text{burial}}$ in the coastal regions of Puerto Rico, rivers discharge onto the shallow submarine shelf, where river POC spreads out and is reworked in the oxygenated environment⁷. It is expected that Icacos and Mameyes have a lower burial efficiency because they have lower SS yields⁸. During extreme events, when over half of biogenic POC export occurs (See Supplementary Table 6), the POC is mainly derived from organic soil and vegetation, but may be converted back to carbon dioxide in the marine environment before deposition (approximately 90% of organic matter in marine deposits is associated with minerals, such as clays and silts⁹). Without measurements available from the ocean adjacent to eastern Puerto Rico, we estimate biogenic $\text{POC}_{\text{burial}}$ using the global river estimates of $22 \pm 5\%$ ¹⁰. The biogenic $\text{POC}_{\text{burial}}$ is estimated at $14.3^{+7.7}_{-6.0}$ and $3.9^{+2.3}_{-1.7}$ $\text{tC km}^{-2} \text{ yr}^{-1}$ for the Icacos and Mameyes, respectively (Table 1; See Supplementary Table 2). The Icacos biogenic $\text{POC}_{\text{burial}}$ yields overlap with the Hokitika and the Whatoroa rivers, in New Zealand, and yields from the Mameyes overlap with the Peel River and the Arctic Red, in Canada, and the Capesterre, in Guadeloupe⁸.”

We have also added this text suggesting future research on the topic on lines 137 to 140:

“To determine the true extent of which the Luquillo catchments are geological carbon sinks, a comprehensive assessment of river biogenic POC burial efficiency in the marine environment is needed.”

Line 122: maybe add a sentence says, the left carbon sink is attributed by the silicate weathering which generate DIC.

We have modified the first sentence now on lines 126 to 128:

“The estimated geological carbon (atmospheric CO_2) sink was $-25.3^{+6.3}_{-7.9}$ and $-17.3^{+2.1}_{-2.6}$ $\text{tC km}^{-2} \text{ yr}^{-1}$ for the Icacos and Mameyes respectively (Table 1), **which comprised of $\text{POC}_{\text{burial}}$ and DIC production from silicate weathering.**”

Line 122: The largest measured carbon sink is in term of yield, not flux. Please clarify it. Same in Line 130, the studied rivers are small, thus it may not suit to say they are important carbon sinks.

You bring up a good point. We were describing the yields (not fluxes) and have modified the text accordingly from lines 128 to 133:

“The largest known carbon sink by area is the Whataroa catchment in New Zealand, $-33 \pm 16 \text{ tC km}^{-2}$ per year⁸; this range, however, overlaps the ranges of the Icacos and Mameyes. The dominant component of the carbon budget in the New Zealand catchment was a large biogenic POC_{burial}, despite a high carbon source contribution from petrogenic POC oxidation⁸. The Capesterre catchment in Guadeloupe is similar to the Icacos and Mameyes, with an estimated carbon sink of $-19 \pm 8 \text{ tC km}^{-2}$ per year^{3,4}.”

Line 135: Please report propagated measurement uncertainty for the POC yields.

We have added the errors on the averages to the yields from Galy et al., 2015 and to our POC yields.

Line 139-140: The bedrock has no biospheric carbon, so I think that the carbon content of bedrock cannot influence the biospheric POC yields.

We have modified this sentence now on lines 151 to 154:

“Two factors contribute to this: the **absence** of carbon content of the bedrock and the greater resilience of igneous rocks in the interior of older uplifts, as compared to more easily eroded marine sediments uplifted at active convergent plate margins³⁷.”

Line 144. How is the value 0.64 calculated? Please clarify it as it is important for assessing the importance role of tropical volcanic mountains on the carbon cycle.

We have explained how we calculated the 0.64. We have also changed how we made the calculation based your previous suggestions. The new text in on lines 169 to 182:

“More broadly speaking, biogenic POC export from rivers draining humid-tropical igneous mountainous islands and island arcs, which encompass a total area of 320,000 square km^{4,12,13}, are also likely underestimated. By assuming that these rivers are similar to those of our study rivers, we estimated biogenic POC fluxes, using the SS yields from Mameyes as the lower bound and from Icacos as the higher bound. We determined the biogenic POC yields using the regression developed by Galy, et al.¹⁴ and then multiplied the area^{4,12,13} to approximate biogenic POC fluxes for this region. Based on the Galy, et al.¹⁴ regression, humid-tropical igneous montane rivers contribute 0.64 to 1.57 MtC yr⁻¹ of biogenic POC fluxes. On the other hand, following the same steps outlined above, but using Eq. 1, this region would contribute fluxes of 5.59 to 23.52 MtC yr⁻¹. This represents a potentially substantial underestimation of biogenic OC export from humid-tropical igneous mountainous islands in the geological carbon cycle^{8,14}. Accordingly, global biogenic OC export, which is currently estimated at 110-230 MtC per year⁸, may be underestimated by 2.1 to 8.7% yr⁻¹. As more research is conducted in these regions, their relative importance to global biogenic POC export will become more apparent.”

Line 151: it is cool that you find the distinct relationships and the cut-off criterion. However, I found that concept between “bedrock petrogenic OC content” with “riverine petrogenic POC

content” is mixed up. I believe that low bedrock petrogenic OC content is always accompanied with low riverine petrogenic POC content. But they are still not equal. Please clarify it.

We have modified the text now on lines 155 to 157:

“There are distinct relationships between biogenic POC relative to SS yield in rivers with petrogenic-carbon-poor bedrock (i.e., **igneous and aluminosilicate metamorphic**) and petrogenic-carbon-rich bedrock (Figure 2a).”

The petrogenic POC content cut-off is almost zero though, because it is calculated as a % of total POC not total SS. So, a petrogenic POC% of 1% of SS is still a petrogenic-carbon-rich river (See Supplementary Table 4).

Line 183: consider re-phrasing “while counting in the carbon sink from the silicate weathering, 44% of the carbon” .

We have removed this text during revisions.

Line 431: a little more detailed summary on the precipitation, erosion, turnover and chemical weathering is needed here..

We have added a more detailed summary on the precipitation and runoff to the main text, now located on line 83 to 85:

“These catchments have very high rainfall (**average of 5,000 and 3,720 mm yr⁻¹ in the Icacos and Mameyes, respectively**)^{17,18} and runoff (**3,906 and 2,770 mm yr⁻¹, respectively**) (Table 1).”

In the methods section we have added in some detail on erosion, turnover, and chemical weathering on lines 563 to 569:

“The sources of suspended sediment erosion from the study catchments are from surficial erosion, bed and bank erosion, and from landslides¹⁹. The total hillslope erosion is estimated to be 750 and 523-2143 t km⁻² yr⁻¹ at the Icacos and Mameyes respectively, where the Mameyes had a particularly high prehistoric landslide erosion (2000 t km⁻² yr⁻¹)¹⁹. Chemical weathering rates of silicate rocks are among the fastest in the world in the Icacos¹¹, with a weathering front of approximately 1 cm per 100 years²⁰. The volcanoclastic rock of the Mameyes weathers event faster, and both catchments have a thick layer of saprolite^{11,21}.”

Line 462: please define the abbreviation LCZO here.

We have defined the abbreviation now as the Luquillo Critical Zone Observatory.

Line 459: please add the information of the sampling frequency and method.

We have added the following information on lines 578 to 590:

“Samples were collected across the hydrograph through a combination of grab sampling (n = 336), depth integrated sampling (n = 272), and event sampling (n = 3391) using a full-sized portable sampler (Teledyne ISCO 6712)³⁸ (See Supplementary Fig. 2 and Table 1). Water samples were filtered for suspended and dissolved load with a 0.2 μm filter membrane³⁸. During the USGS WEBB program, far more samples of most dissolved and suspended constituents were collected than in other studies. Most importantly, SS samples analyzed for biogenic POC were collected during extreme rainfall events where runoff was > 10 mm hr⁻¹ at both Icacos (n = 1) and Mameyes (n = 7); the number of samples analyzed for SS (used to estimate POC and PN, see below) and comprehensive chemistry was considerably greater (See Supplementary Table 1). There have only been two other rivers where SS samples analysed for biogenic POC were collected at extremely high runoff, specifically 11.2 mm hr⁻¹ in Taiwan¹ and 11.1 mm hr⁻¹ in the Erlenbach in the Swiss Alps².”

Line 508-514: consider re-phrasing “The POC contribution to LOI was calculated as 50%”. In addition, what your aim for this paragraph? It is not clear.

We have reworked this paragraph and described why this was an important step. We have modified the text from lines 654 to 659:

“As a check on the POC estimated concentrations, we assessed the maximum possible POC concentration in the SS sample based on LOI, as thousands of SS samples were assessed for LOI. The organic matter contribution to LOI was calculated as twice POC. Whenever twice POC exceeded LOI or estimated LOI (see Stallard⁶ which has the equations for estimated LOI using SS), POC was set to half the LOI. A handful of samples with estimated POC were so capped and the maximum LOI estimated POC concentration was used instead.”

Figure 1: it seems that the samples with the largest suspended sediment concentration are not included in the POC% analysis. Will it influence the flux calculation?

The samples at high SSC were combusted for LOI, and we measured the available samples for POC% (the colored circles). We estimated the POC% for all of the SS samples used in the LOADEST load estimations. The details of this estimation are now on lines 648 to 653:

“To extend the effective range along the hydrograph, POC and PN concentrations were estimated for SS samples without POC or PN measurements (Figure 1; See Supplementary Table 1 and Table 8). Constituent concentrations (mg L⁻¹) were log-log transformed and correlation coefficients regressions were generated. POC concentrations (mg L⁻¹) were estimated as:

$$POC_{est\ Icacos} = 0.054 \times SS^{0.977}, r^2 = 0.96 \quad (Eq. 5)$$

$$POC_{est\ Mameyes} = 0.069 \times SS^{0.977}, r^2 = 0.91 \quad (Eq. 6)”$$

We have added text to the Figure 1 caption:

“Biogenic POC (mg L^{-1}) was estimated from SS samples that were not measured for POC% (grey circles) using Eq. 5 for Icacos and Eq. 6 for Mameyes.”

Figure 2: please use a different symbol for the River Icacos and Mameyes.

Done. We've filled in the symbols with a different color for each river, matching Figure 1.

Figure S2: the symbol is not very clear. Please consider using colour symbols.

Done.

Jin Wang,
Institute of Earth Environment, Chinese Academy of Science.

References

- 1 Kao, S. J. *et al.* Preservation of terrestrial organic carbon in marine sediments offshore Taiwan: mountain building and atmospheric carbon dioxide sequestration. *Earth Surface Dynamics* **2**, 127-139 (2014).
- 2 Smith, J. C. *et al.* Runoff-driven export of particulate organic carbon from soil in temperate forested uplands. *Earth and Planetary Science Letters* **365**, 198-208 (2013).
- 3 Lloret, E. *et al.* Comparison of dissolved inorganic and organic carbon yields and fluxes in the watersheds of tropical volcanic islands, examples from Guadeloupe (French West Indies). *Chemical Geology* **280**, 65-78, (2011).
- 4 Lloret, E. *et al.* Dynamic of particulate and dissolved organic carbon in small volcanic mountainous tropical watersheds. *Chemical Geology* **351**, 229-244 (2013).
- 5 Strauch, A. M., MacKenzie, R. A., Giardina, C. P. & Bruland, G. L. Influence of declining mean annual rainfall on the behavior and yield of sediment and particulate organic carbon from tropical watersheds. *Geomorphology* **306**, 28-39, (2018).
- 6 Stallard, R. F. Data processing and computation to characterise hydrology and compare water quality of four watersheds in Eastern Puerto Rico, Appendix 1 in *Water quality and landscape processes of four watersheds in eastern Puerto Rico: U.S. Geological Survey Professional Paper 1789* (eds Murphy, S. F. & Stallard, R. F.) 263-287 (U. S. Geological Survey, Reston, VA, 2012).
- 7 Warne, A. G., Webb, R. M. & Larsen, M. C. Water, sediment, and nutrient discharge characteristics of rivers in Puerto Rico, and their potential influence on coral reefs. 58 (US Department of the Interior, US Geological Survey, 2005).
- 8 Hilton, R. G. & West, A. J. Mountains, erosion and the carbon cycle. *Nature Reviews Earth & Environment* **1**, 284-299, (2020).
- 9 Hedges, J. I. & Keil, R. G. Sedimentary organic-matter preservation - an assessment and speculative synthesis. *Marine Chemistry* **49**, 81-115, (1995).
- 10 Burdige, D. J. Burial of terrestrial organic matter in marine sediments: A re-assessment. *Global Biogeochemical Cycles* **19**, (2005).
- 11 White, A. F. *et al.* Chemical Weathering in a Tropical Watershed, Luquillo Mountains, Puerto Rico: I. Long-Term Versus Short-Term Weathering Fluxes. *Geochimica et Cosmochimica Acta* **62**, 209-226 (1998).

- 12 Allègre, C. J. *et al.* The fundamental role of island arc weathering in the oceanic Sr isotope budget. *Earth and Planetary Science Letters* **292**, 51-56, (2010).
- 13 Dessert, C., Dupré, B., Gaillardet, J., François, L. M. & Allegre, C. J. Basalt weathering laws and the impact of basalt weathering on the global carbon cycle. *Chemical Geology* **202**, 257-273, (2003).
- 14 Galy, V., Peucker-Ehrenbrink, B. & Eglinton, T. Global carbon export from the terrestrial biosphere controlled by erosion. *Nature* **521**, 204-207 (2015).
- 15 Stallard, R. F. Weathering, landscape equilibrium, and carbon in four watersheds in eastern Puerto Rico, Ch. H in *Water quality and landscape processes of four watersheds in eastern Puerto Rico: U.S. Geological Survey Professional Paper 1789* (eds Murphy, S. F. & Stallard, R. F.) 199-248 (US Geological Survey, Reston, VA, 2012).
- 16 Murphy, S. F., Stallard, R. F., Larsen, M. C. & A., G. W. Physiography, Geology, and Land Cover of Four Watersheds in Eastern Puerto Rico, Ch. A in *Water quality and landscape processes of four watersheds in eastern Puerto Rico: U.S. Geological Survey Professional Paper 1789* (eds Murphy, S. F. & Stallard, R. F.) 3-23 (U.S. Geological Survey Professional Paper 1789, Reston, VA, 2012).
- 17 Wlostowski, A. N. *et al.* Signatures of Hydrologic Function Across the Critical Zone Observatory Network. *Water Resources Research* **57**, e2019WR026635, (2021).
- 18 Murphy, S. F., Stallard, R. F., Scholl, M. A., González, G. & Torres-Sánchez, A. J. Reassessing rainfall in the Luquillo Mountains, Puerto Rico: Local and global ecohydrological implications. *PLOS ONE* **12**, e0180987, (2017).
- 19 Larsen, M. C. Landslides and sediment budgets in four watersheds in eastern Puerto Rico: Ch. F in *Water quality and landscape processes of four watersheds in eastern Puerto Rico: U.S. Geological Survey Professional Paper 1789* (eds Murphy, S. F. & Stallard, R. F.) Ch. F, 153-178 (U. S Geological Survey, 2012).
- 20 Brown, E. T., Stallard, R. F., Larsen, M. C., Raisbeck, G. M. & Yiou, F. Denudation rates determined from the accumulation of in situ-produced ^{10}Be in the luquillo experimental forest, Puerto Rico. *Earth and Planetary Science Letters* **129**, 193-202, (1995).
- 21 Buss, H. L. & White, A. F. Weathering Processes in the Icacos and Mameyes Watersheds in Eastern Puerto Rico, Ch. I in *Water quality and landscape processes of four watersheds in eastern Puerto Rico: U.S. Geological Survey Professional Paper 1789* (eds Murphy, S. F. & Stallard, R. F.) (U.S. Geological Survey, Reston, VA, USA, 2012).

- 22 Ludwig, W., Probst, J.-L. & Kempe, S. Predicting the oceanic input of organic carbon by continental erosion. *Global Biogeochemical Cycles* **10**, 23-41, (1996).
- 23 Coynel, A., Seyler, P., Etcheber, H., Meybeck, M. & Orange, D. Spatial and seasonal dynamics of total suspended sediment and organic carbon species in the Congo River. *Global Biogeochemical Cycles* **19**, (2005).
- 24 Stein, R., Macdonald, R. W., Stein, R. & MacDonald, R. W. *The organic carbon cycle in the Arctic Ocean*. (2004).
- 25 Ingri, J., Widerlund, A. & Land, M. Geochemistry of Major Elements in a Pristine Boreal River System; Hydrological Compartments and Flow Paths. *Aquatic Geochemistry* **11**, 57-88, (2005).
- 26 Leithold, E. L., Blair, N. E. & Perkey, D. W. Geomorphologic controls on the age of particulate organic carbon from small mountainous and upland rivers. *Global Biogeochemical Cycles* **20** (2006).
- 27 Baedecker, M. J. *Water, energy, and biogeochemical budgets: A watershed research program*. Vol. 165 (US Department of the Interior, US Geological Survey, 2000).
- 28 Larsen, M. C., Collar, P. D. & Stallard, R. F. Research plan for the investigation of water, energy, and biogeochemical budgets in the Luquillo Mountains, Puerto Rico. 19 (U.S. Geological Survey, 1993).
- 29 Clark, K. E. *et al.* Tropical river sediment and solute dynamics in storms during an extreme drought. *Water Resources Research* (2017).
- 30 Hilton, R. G. Climate regulates the erosional carbon export from the terrestrial biosphere. *Geomorphology*, 118-132 (2017).
- 31 Torres, M. A., West, A. J. & Li, G. Sulphide oxidation and carbonate dissolution as a source of CO₂ over geological timescales. *Nature* **507**, 346-349, (2014).
- 32 U.S. Geological Survey. USGS 50065500 Rio Mameyes Nr Sabana, PR, in USGS water data for the Nation: U.S. Geological Survey National Water Information System database. Accessed February 15, 2016, at <https://doi.org/10.5066/F7P55KJN>. [Site information directly accessible at http://waterdata.usgs.gov/pr/nwis/uv?site_no=50065500.], (2016).
- 33 U.S. Geological Survey. USGS 50075000 Rio Icacos Nr Naguabo, PR, in USGS water data for the Nation: U.S. Geological Survey National Water Information System database. Accessed February

- 15, 2016, at <https://doi.org/10.5066/F7P55KJN>. [Site information directly accessible at http://waterdata.usgs.gov/pr/nwis/uv?site_no=50075000.], (2016).
- 34 Runkel, R. L., Crawford, C. G. & Cohn, T. A. Load estimator (LOADEST): a FORTRAN program for estimating constituent loads in streams and rivers. Report No. 4-A5, (2004).
- 35 McDowell, W. H. & Asbury, C. E. Export of carbon, nitrogen, and major ions from three tropical montane watersheds. *Limnology and Oceanography* **39**, 111-125 (1994).
- 36 Stallard, R. F. Terrestrial sedimentation and the carbon cycle: Coupling weathering and erosion to carbon burial. *Global Biogeochemical Cycles* **12**, 231-257 (1998).
- 37 Stallard, R. F. Weathering and erosion in the humid tropics in *Physical and chemical weathering in geochemical cycles* (eds Lerman, A. & Meybeck, M.) 225-246 (Kluwer Academic Publishers, Dordrecht, Netherlands, 1988).
- 38 Murphy, S. F. & Stallard, R. F. Methods used to analyse water quality of four watersheds in Eastern Puerto Rico, Appendix 2 in *Water quality and landscape processes of four watersheds in eastern Puerto Rico: U.S. Geological Survey Professional Paper 1789* (eds Murphy, S. F. & Stallard, R. F.) 289-292 (U. S. Geological Survey, Reston, VA, 2012).

REVIEWERS' COMMENTS

Reviewer #2 (Remarks to the Author):

This revised manuscript is well improved. The authors have done a good job of addressing my previous concerns. I recommend publication of this manuscript in Nature Communications. I only have two minor comments for your consideration.

1. I found the “biospheric POC” has been replaced by “biogenic POC” in the revised manuscript. It is OK. Please also change the biospheric in Line 50 and 53. I also suggest that it is better to define the “biogenic POC” at it first appearance.

2. Line 165 : consider removing “, where Y_{ss} represents the annual SS yield” to the front of Line 167.

Author Response to Reviewer Comments

Response to Editor and Reviewers' Comments for Nature Communications manuscript ID: NCOMMS-21-27380A

Comments are copied below with our responses in green text, manuscript text is in quotes with small additions to quoted text being bolded.

REVIEWERS' COMMENTS

Reviewer #2 (Remarks to the Author):

This revised manuscript is well improved. The authors have done a good job of addressing my previous concerns. I recommend publication of this manuscript in Nature Communications. I only have two minor comments for your consideration.

1. I found the “biospheric POC” has been replaced by “biogenic POC” in the revised manuscript. It is OK. Please also change the biospheric in Line 50 and 53. I also suggest that it is better to define the “biogenic POC” at it first appearance.

We have changed biospheric to biogenic POC on lines 48 and 50. We have defined biogenic POC on lines 43: “There is a drawdown of atmospheric CO₂ when river particulate organic carbon (POC) from the biosphere (**biogenic POC**) is buried...”

2. Line 165 : consider removing “, where Y_{ss} represents the annual SS yield” to the front of Line 167.

We have moved this text to the front of line 167, now line 164.